# Regime shift in Arctic Ocean sea ice thickness

Hiroshi Sumata[1✉], Laura de Steur[1], Dmitry V. Divine[1], Mats A. Granskog[1] & Sebastian Gerland[1]

Manifestations of climate change are often shown as gradual changes in physical or biogeochemical properties[1]. Components of the climate system, however, can show stepwise shifts from one regime to another, as a nonlinear response of the system to a changing forcing[2]. Here we show that the Arctic sea ice regime shifted in 2007 from thicker and deformed to thinner and more uniform ice cover. Continuous sea ice monitoring in the Fram Strait over the last three decades revealed the shift. After the shift, the fraction of thick and deformed ice dropped by half and has not recovered to date. The timing of the shift was preceded by a two-step reduction in residence time of sea ice in the Arctic Basin, initiated first in 2005 and followed by 2007. We demonstrate that a simple model describing the stochastic process of dynamic sea ice thickening explains the observed ice thickness changes as a result of the reduced residence time. Our study highlights the long-lasting impact of climate change on the Arctic sea ice through reduced residence time and its connection to the coupled ocean–sea ice processes in the adjacent marginal seas and shelves of the Arctic Ocean.

The extent of Arctic sea ice exhibited negative trends both in summer and winter over the last three decades[1,3]. Such a retreat in summer sea ice results in an increase of open water areas and prolongation of ice-free conditions in the adjacent marginal seas and shelves of the Arctic Ocean[4,5]. The adjacent, ice-free seas absorb more solar energy during summer and store heat in the upper ocean[6]. The heat enhances ice melt in the marginal ice zone in summer and delays the onset of new ice formation in the autumn[7]. These processes influence Arctic-wide sea ice properties through large-scale sea ice motion carrying the ice from the marginal seas to the central Arctic, a process known as Transpolar Drift (TPD) Stream[8].

A large fraction of the Arctic Ocean's sea ice is formed in the marginal seas of the Arctic Ocean, from the Alaskan to Siberian sectors of the Arctic[9], that is, the Beaufort, Chukchi, East Siberian and Laptev seas. It is then transported across the Arctic to sustain the perennial ice pack (Fig. 1a). A main part of sea ice formed in the Alaskan sector circulates in the Canada Basin[10], while a part of the ice is pushed towards the Siberian sector via the Chukchi Sea[11]. Sea ice formed in the Siberian sector, together with the ice from the Alaskan sector, joins the TPD emanating from offshore of the Siberian shelves. The TPD transports the ice to the central Arctic and further towards the Atlantic sector of the Arctic[8,12]. Finally, it exits the Arctic Ocean through the Fram Strait, located between north-eastern Greenland and Svalbard. Sea ice motion and ice age estimates from observations indicate an acceleration of ice motion and hence decrease of residence time of sea ice in the Arctic Basin in recent decades[13,14].

Because up to approximately 90% of sea ice outflow from the Arctic Ocean to the Subarctic North Atlantic occurs through the Fram Strait[15,16], sea ice properties observed in the Fram Strait represent basin-wide characteristics of Arctic sea ice[14,17]. The Fram Strait Arctic Outflow Observatory has been monitoring sea ice and ocean properties in the core of the outflow at a latitude of approximately 79° N[18] (marked red in Fig. 1a)

since 1990. The observatory has provided a unique, near-continuous time series of sea ice thickness for the last three decades[19] (Methods).

In this study, we show that in 2007, a regime shift of sea ice thickness occurred in the Fram Strait. The shift was characterized by an abrupt reduction of deformed thick ice (a 52% reduction of sea ice thicker than 4 m) and an increased uniformity of ice thickness distribution (height of the modal peak increased by 67%). The timing of the shift follows a drop in Arctic-wide sea ice residence time in 2005 and 2007. We introduce a simple model describing the stochastic process of dynamic ice thickening and explain the relationship between the changes in Arctic sea ice residence time and thickness distribution.

## Changes in ice thickness distribution

Sea ice observed in the Fram Strait consists of a variety of ice types and thicknesses, reflecting its thermal and dynamic history before exiting the Arctic Basin[20,21]. Typically, ice thickness distribution in the Fram Strait is bimodal (Fig. 1b and Extended Data Fig. 5). The first peak represents thin sea ice (less than 0.5 m) formed in the vicinity of the Fram Strait, while the second peak (that is, the modal peak), which has a 1.5–3 m thickness, primarily represents thermodynamically grown sea ice that has been transported across the Arctic Basin. Finally, the distribution has a long tail towards thicker ice fractions, corresponding to dynamically thickened sea ice (from thinner thermodynamically grown sea ice) due to ridging and rafting. The second peak and tail can be reasonably well approximated using a log-normal function (Fig. 1b, Extended Data Fig. 5 and Methods). This part of the distribution represents sea ice properties in the central Arctic and gives information on combined thermal and dynamic forcing on sea ice from the time of its formation to its arrival in the Fram Strait.

The distribution of sea ice thickness has changed substantially in the last three decades, reflecting the environmental changes in the Arctic

[1]Norwegian Polar Institute, Fram Centre, Tromsø, Norway. ✉e-mail: hiroshi.sumata@npolar.no

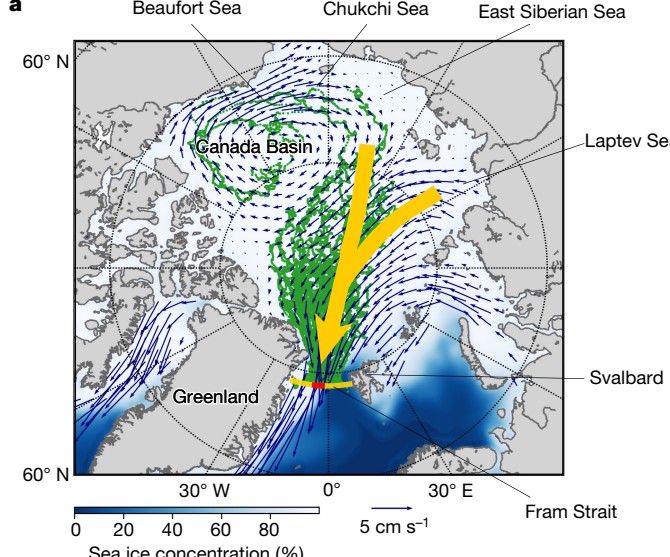

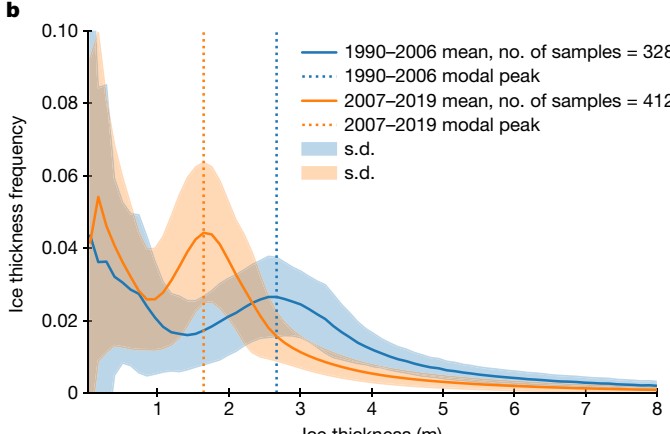

**Fig. 1 | Map of Arctic Ocean and sea ice thickness distribution in the Fram Strait. a**, Arctic Ocean and its marginal seas, with winter sea ice concentration (1980–2018 mean, white-blue shading, calculated from OSI SAF[51]), ice drift field (blue arrows, Polar Pathfinder Daily 25 km EASE-Grid Sea Ice Motion Vectors v.4.1)[52], 83 ice-tethered buoy tracks that arrived in the Fram Strait (green lines) and TPD Stream (yellow shade). The buoy tracks were obtained from the International Arctic Buoy Programme[53]. The Fram Strait Arctic Outflow Observatory is shown by the red bar. **b**, Mean sea ice thickness distribution in the Fram Strait before and after 2007. The distributions were derived on a monthly basis by all available ULS data from 1990 to 2019 (described in the Methods) and averaged across two periods: 1990–2006 and 2007–2019. The Matplotlib basemap toolkit was used to plot the map.

and consequent reduction of mean ice thickness in the Arctic Ocean[21]. In the Fram Strait, the thickness of the modal peak has been reduced by approximately 1 m (2.7 m to 1.7 m; Fig. 1b, dashed lines); a fraction of the ice in the mode (height of the peak) has increased by 67% from 1990–2006 to 2007–2019 (Fig. 1b, blue versus orange line). The tail of the distribution, corresponding to the deformed fraction of ice, has also substantially changed between the two periods: the thickness distribution for 1990–2006 (Fig. 1b, blue line) was characterized by thicker modal thickness with a smaller and broader modal peak and a larger fraction of deformed ice (thick and deformed ice regime), while the period after 2007 (Fig. 1b, orange line) is characterized by a thinner modal thickness with more compact distribution of ice thickness around the mode and a smaller fraction of deformed ice (thin and more uniform ice regime). These findings from the mooring observations are consistent with in situ observations[22].

Figure 2a shows the time series of ice thickness distribution in the Fram Strait for the last three decades. Darker shading indicates a larger fraction of sea ice at the corresponding thickness. A zonal band with a darker shade ranging approximately from 3.0 m to 1.5 m, depicts the modal thickness of multi-year sea ice. The temporal variation of this band describes the long-term changes of modal sea ice thickness in the central Arctic. The time series clearly shows that the change of the thickness distribution (Fig. 1b) has not been a gradual process but that a distinct shift from thick and deformed ice regime to thinner and more uniform ice regime occurred around 2007 (visualized by the intensity of the shade at the modal peak in Fig. 2a). The shift is further evidenced by coincident changes in modal peak height (Fig. 2b) and the variance of ice thickness distribution (Fig. 2c). The modal peak height and variance were obtained from a fitted log-normal function to each distribution (Methods). The modal peak height gives a measure of compactness of the distribution around the mode, while the variance gives an indication of deformed fraction of sea ice relative to the modal peak. The sequential *t*-test analysis of regime shifts[23] we applied to these time series detected a shift in both peak height and variance in 2007 (Extended Data Fig. 6 and Extended Data Table 1). This indicates that until 2007, ice floes consisted of sea ice with a variety of thicknesses towards thicker ice, whereas from 2007, they have consisted of ice of more uniform thickness with a smaller deformed fraction. This is also clearly visible in shifts with a 1-year delay in the fraction of thick ice (that is, ice thickness exceeding 4 and 5 m) shown in Fig. 2d. This fraction corresponds to ice thicker than the maximum theoretical thickness of thermodynamically grown ice derived by different sea ice models[24,25], that is, giving a measure of the fraction of dynamically deformed ice. The shift of ice thickness distribution in 2007 and 2008 indicates that a sudden reduction of dynamic forcing on the ice occurred at that time.

## Reduced residence time of Arctic sea ice

Figure 3a shows a time series of mean residence time of sea ice floes in the Arctic Ocean that arrived in the Fram Strait (Methods). The timing of the observed shift in thickness distribution coincided with the timing of a reduction in residence time. The mean residence time showed a two-step shift from 4.3 years to 2.7 years in 2005 and 2007 (detected by sequential *t*-test analysis of regime shifts; Methods). A coincident exceptional long residence time (Fig. 3a) and large variance of thickness distribution (Fig. 2c) occurred in 2017, reinforcing the connection between residence time and thickness distribution. Although the mean residence time dropped in 2005 and 2007, the area of ice formation (coloured dots in Fig. 3b,c) and ice trajectories across the Arctic basins (grey cloud in Fig. 3b,c) have not changed notably. The slight offshore and westward shift of ice formation areas[26] is not sufficient to explain the reduction of residence time by approximately 1.6 years (Fig. 3a and colour of the dots in Fig. 3b,c).

The reduction of summer sea ice concentration in areas of sea ice formation, on the other hand, correlates with the reduction in residence time ($r = 0.65$ in the Alaskan sector, 0.73 in the Siberian sector, ice concentration leads 1 year). Figure 4a shows the difference in September sea ice concentration between two periods, 1990–2006 and 2007–2019. Areas that show the largest decrease coincide with areas of sea ice formation (blue shading in Fig. 4a versus coloured dots in Fig. 3b,c); a reduction of sea ice concentration occurred both in the Alaskan and Siberian sectors in 2005 and 2007 (Fig. 4c and Extended Data Table 1). The September mean ice concentration dropped from 46% to 26% in the Alaskan sector and from 57% to 26% in the Siberian sector and has not recovered to date (Fig. 4c). Concurrently, the September mean sea surface temperature in these areas has risen from below 0 °C to 0.6 °C (Extended Data Fig. 1). These changes make it difficult for ice formed during a previous winter to survive the summer melt in these areas and survive into the following year. This is manifested in the drop of residence time of ice floes in areas of sea ice formation (Extended

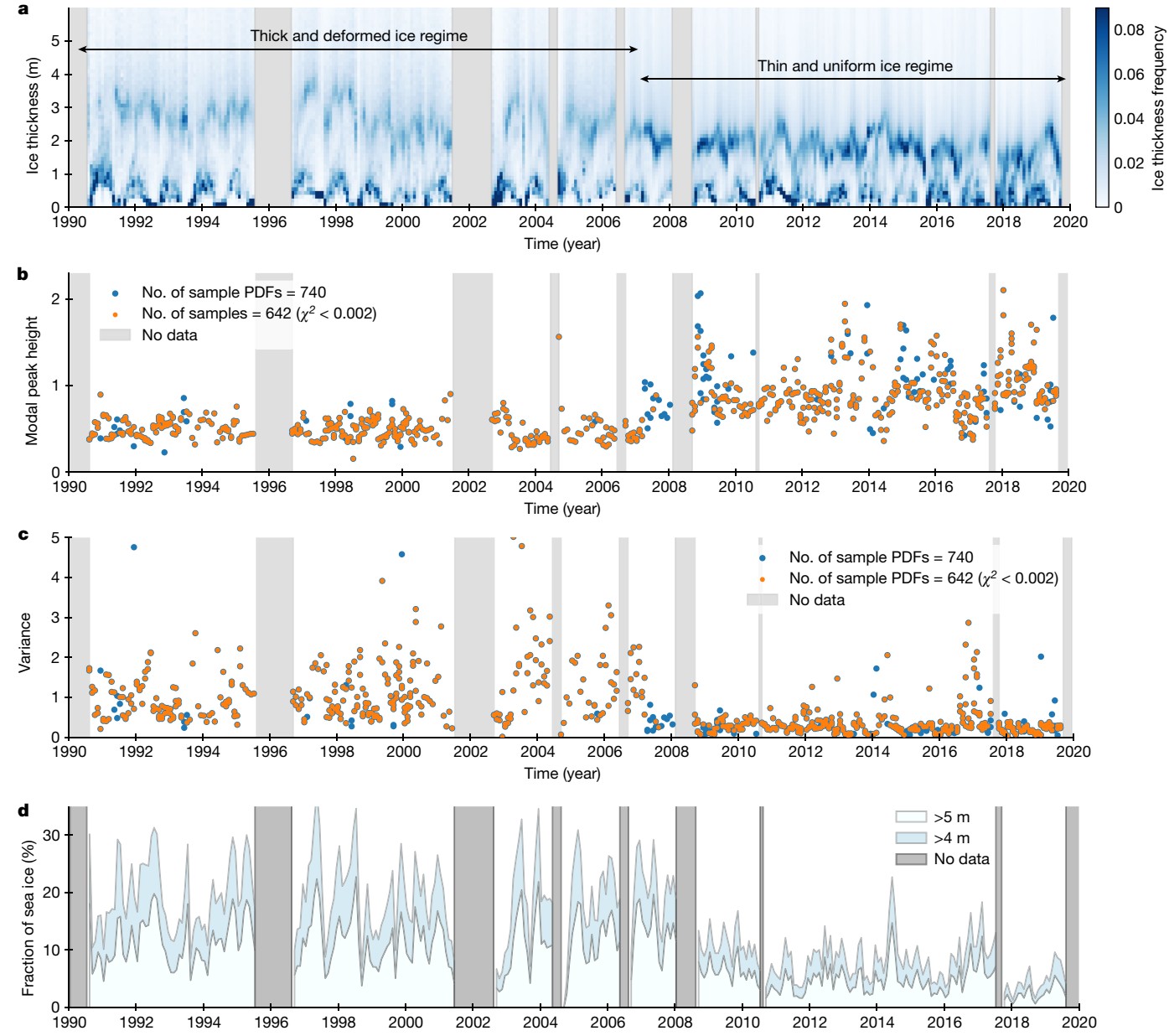

**Fig. 2 | Sea ice thickness properties observed in the Fram Strait in the last three decades. a–d**, Time series of sea ice thickness distribution (**a**), modal peak height (**b**) and variance of ice thickness distributions (**c**), and fraction of sea ice thicker than two thresholds, that is, 4 m and 5 m, respectively (**d**). In

**b**,**c**, $\chi^2$ is a sum of the squared residuals at each log-normal function fitting. Derivations of the ice thickness distribution, modal peak height and variance are described in the Methods.

Data Fig. 2). The mean residence time of ice floes in the Siberian sector that arrived at the Fram Strait after 2007, was reduced from 15 to 6 months, indicating that most of the ice floes cannot survive the summer melt season and only new ice floes that formed after the summer enter the TPD.

Acceleration of ice drift speed in the Arctic Ocean[27,28] has also contributed to the shorter residence time of sea ice. Figure 4b shows the difference in sea ice motion between the two periods, 1990–2006 and 2007–2019. Enhanced anticyclonic ice motion in the Canada Basin and acceleration of the TPD are clearly visible. These changes also exhibited a clear shift in 2007 (Fig. 4d and Extended Data Table 1). Westward ice drift speed in the Alaskan sector increased by 71% (from 2.0 cm s$^{-1}$ to 3.5 cm s$^{-1}$), northward ice drift in the Siberian sector increased by 42% (from 1.2 cm s$^{-1}$ to 1.7 cm s$^{-1}$) and the TPD accelerated by 37% (from 2.3 cm s$^{-1}$ to 3.2 cm s$^{-1}$), respectively. The acceleration of the TPD

shortened the residence time of sea ice in the central Arctic Ocean by about four months on average. Concurrent shorter residence times in the ice formation areas and acceleration of the TPD led to a reduction of the mean residence time of sea ice by 1.6 years (from 4.3 to 2.7 years) after 2007.

## Shorter residence time and thinner ice

The relationship between the observed regime shift (Figs. 1b and 2) and reduced residence time (Fig. 3) is explained by a dynamic ice thickening process. Heat loss in open water areas to the atmosphere in autumn, followed by continued cooling in winter, forms a uniform ice thickness distribution with a high modal peak[24] (thermodynamic forcing is governed by synoptic scales of $O(10^3)$ km). Dynamic forcing, on the other hand, increases the fraction of deformed thick ice forming the

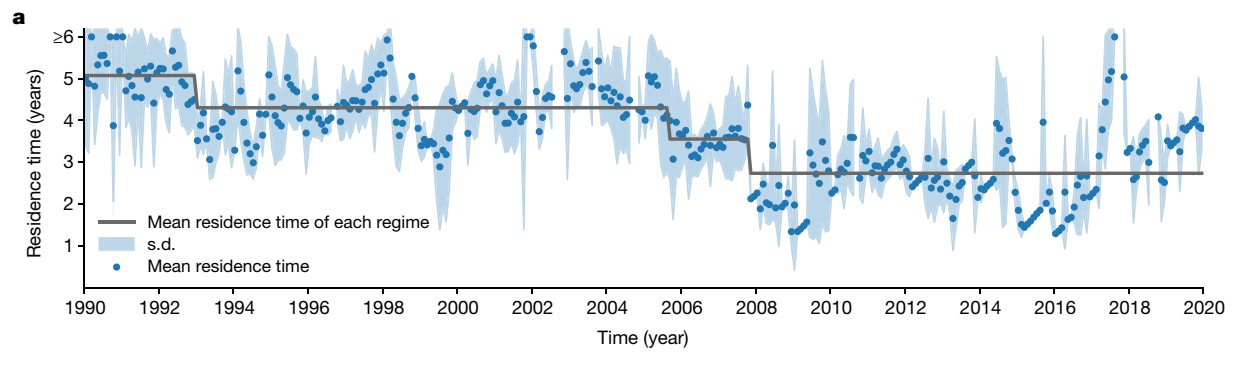

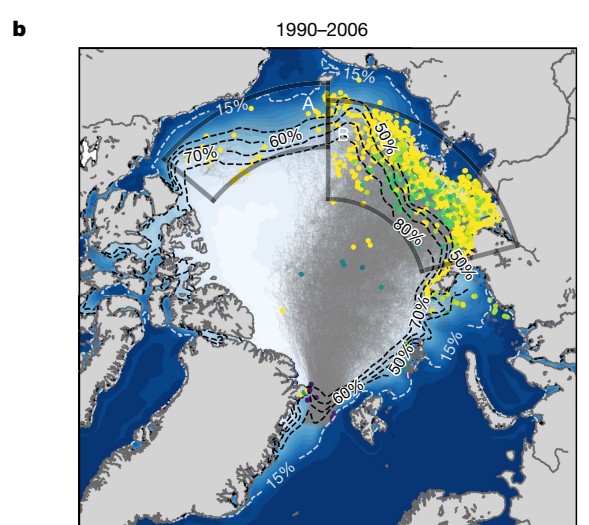

**Fig. 3 | Residence time and origins of sea ice in the Arctic Ocean that reached the Fram Strait. a–c**, Time series of residence time of ice floes in the Arctic Basin (**a**) and origins and pathways of ice floes before (**b**) and after 2007 (**c**). **a**, The abscissa references the time of arrival in the Fram Strait. The grey line shows the mean residence time in each regime detected by sequential *t*-test analysis of regime shifts. **b,c**, The location of the dots depicts areas of sea ice formation, while the colour of the dots indicates the time of sea ice formation relative to their arrival in the Fram Strait. The grey clouds in **b** and **c** show the trajectories of ice floes from their origins to the Fram Strait. The background

colour (navy–white shading) in **b** and **c** shows the mean sea ice concentration in September for the corresponding periods (OSI SAF[51]). The contours of the mean sea ice concentration are represented by the dashed lines (80%, 70%, 60% and 50% contours shown in black and 15% contour shown in white). Two polygons indicate the sea ice formation area in the Alaskan (A) and Siberian (B) sectors: the time series of sea ice concentration and ice drift speed in these areas are shown in Fig. 4. See the Methods for details of the backward trajectory calculation and residence time estimates. The Matplotlib basemap toolkit was used to plot the map.

tail of the thickness distribution. Divergence, convergence and shear of wind and ocean currents cause mechanical fracturing of ice floes. When the forcing is convergent and once the local internal stress in the ice pack has exceeded the threshold, dynamic ice thickening occurs by ridging and/or rafting[25,29].

Dynamic ice thickening is a stochastic process because the deformation occurs in a small fraction of ice while the rest of the ice is unchanged when a dynamic event occurs. Ice thickness gained by an event (that is, ridging or rafting) also varies in space and differs between events. Another characteristic of dynamic thickening is its dependence on ice thickness. Thicker ice has a larger potential to get thicker as it can exert stronger compressive force on the ice forming ridges and rafting[30,31]. These characteristics enable us to formulate the process by a proportionate ice thickening of stochastic ice thickness, $X$, as,

$$X_i = a_{i-1} X_{i-1} \tag{1}$$

where $i$ denotes the time index counting dynamic ice thickening events and $a_{i-1}$ is the proportionate thickening increment due to the event at $i-1$. The increment represents the stochasticity of the dynamic thickening process. After $m$-times dynamic growth events, stochastic ice thickness is given by:

$$X_m = X_0 \Pi_{i=0}^{i=m-1} a_i \tag{2}$$

where $X_0$ is the initial ice thickness, that is, sea ice without dynamic thickening, and $\Pi$ is a product operator. Taking the natural logarithm of equation (2) gives:

$$\ln X_m = \ln X_0 + \ln a_{m-1} + \ln a_{m-2} + ... + \ln a_0 \tag{3}$$

As $\ln X_0$, $\ln a_0$, $\ln a_1$, ..., and $\ln a_i$ are uncorrelated (the thickening at each dynamic event is a stochastic process), the probability function of $\ln X_m$ gives a normal distribution for large $m$ (central limit theorem[32]), hence the probability function of $X_m$ gives a log-normal distribution (Gibrat's law[33]). The shape of the distribution varies with $m$, that is, the number of dynamic growth events. The larger the $m$, the larger the variance. If we assume that the number of dynamic growth events $m$ is proportional to the residence time of ice floes in the Arctic Ocean on annual timescales, a longer residence time leads to larger variance, that is, a larger fraction of deformed ice. As the number of such events has not considerably changed in the last three decades[34], $m$ can be linearly related to the residence time of sea ice in the Arctic Basin.

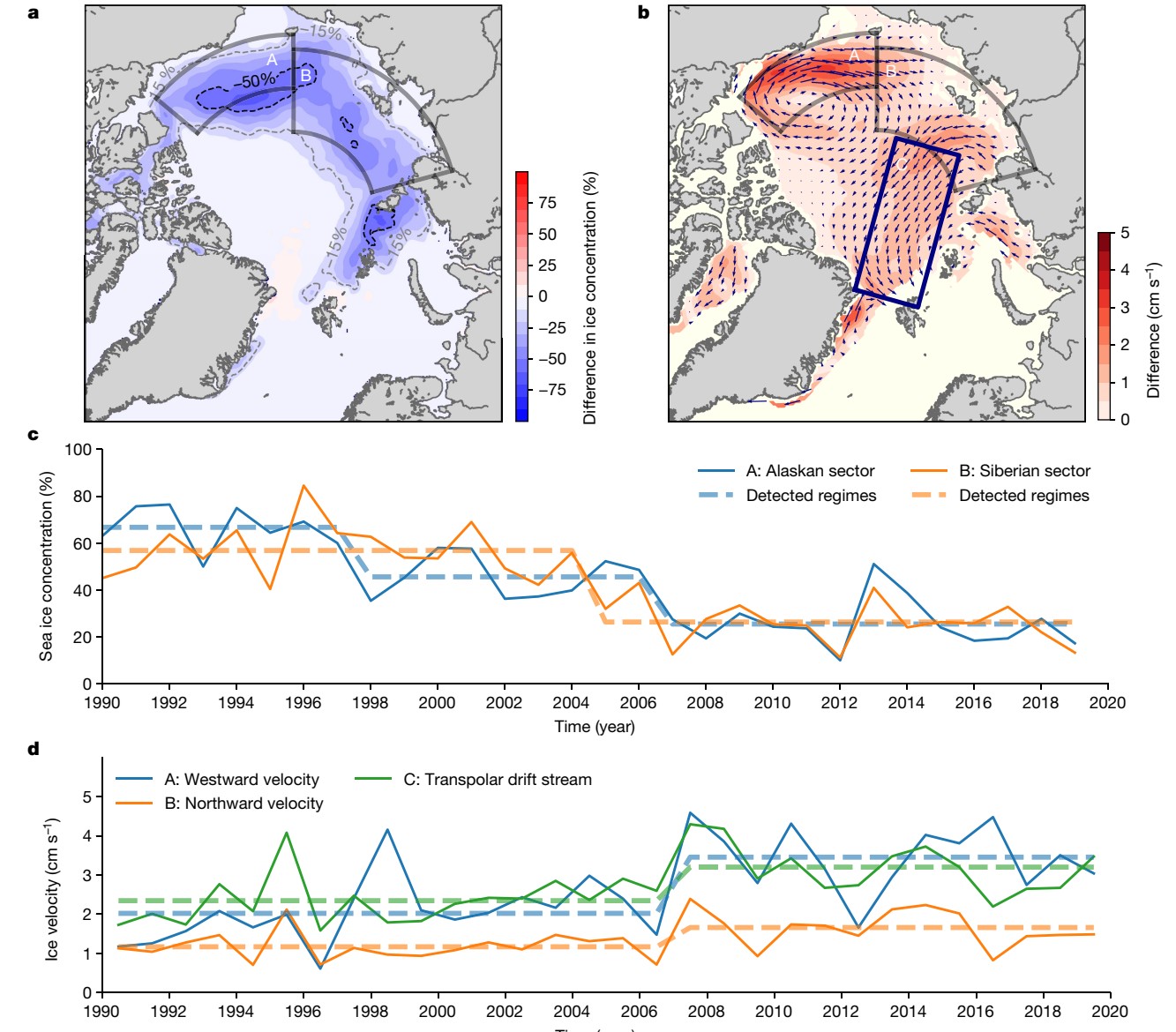

**Fig. 4 | Changes of sea ice concentration and sea ice motion. a,b,** Difference of September sea ice concentration (**a**) and ice drift speed (**b**) between the two periods: 1990–2006 and 2007–2019. **c,d,** Time series of mean sea ice concentration in September (**c**) and mean sea ice drift speed in selected regions (**d**). **a,** The positive (negative) values indicate increase (decrease) in the latter period. **b,** The difference in sea ice drift vector is shown by the arrows, while its magnitude is shown by the colour. The difference in sea ice drift field in **b** was calculated from ice drift vectors from December to May. The time series in **c** are the areal average of the Alaskan (A) and Siberian sectors (B) shown by the solid black polygon in **a**, while those in **d** are the areal average of A,

B and C: the TPD Stream is shown by the rectangular box labelled C. The ice drift speed of the TPD in **d** shows the annual mean ice drift speed in box C (vector component parallel to the main axis of box C, positive value oriented to the Fram Strait), whereas those in A and B are calculated without three summer months (August to October) to exclude under-represented ice motion due to very low spatial coverage in recent years. **c,d,** The dashed lines indicate the detected regimes (Extended Data Table 1). Sea ice concentration from OSI SAF[51] and sea ice drift from Polar Pathfinder Daily 25 km EASE-Grid Sea Ice Motion Vectors v.4.1 (ref.[52]) were used to derive the variables. The Matplotlib basemap toolkit was used to plot the map.

Figure 5 shows examples of the probability density function of $X_m$ for different $m$ (Methods). The plot demonstrates that the shorter residence (smaller $m$) gives higher modal peak and less deformed sea ice. The simple proportionate thickening process conceptually explains (1) the shape of the observed ice thickness distribution (the log-normal shapes in Fig. 1b) and, more importantly, (2) the reduction of the fraction of deformed ice for the shorter residence time after 2007 (Fig. 2).

## Discussion

The relationship between sea ice residence time in the Arctic Ocean and ice thickness distribution highlights the importance of coupled

ocean–sea ice processes in the Alaskan and Siberian sectors of the Arctic (areas A and B in Fig. 3b). Several interrelated factors have become more prominent in the late twentieth century and have contributed to preconditioning the ocean–sea ice system before the stepwise changes in the ice formation areas: Arctic-wide rise of surface air temperature[35], thinning of sea ice[36], decrease of sea ice albedo[37] concurrent with a reduction of multi-year sea ice[38], increase of ocean heat flux through the Bering Strait[39] and increase of the upper ocean heat content[40]. September sea ice concentration in the Siberian sector dropped below 40% in 2005 and the dramatic Arctic summer sea ice extent minimum occurred in 2007 (ref.[41]). This series of events initiated intensive and widespread ice–albedo feedback in the Alaskan and Siberian sectors

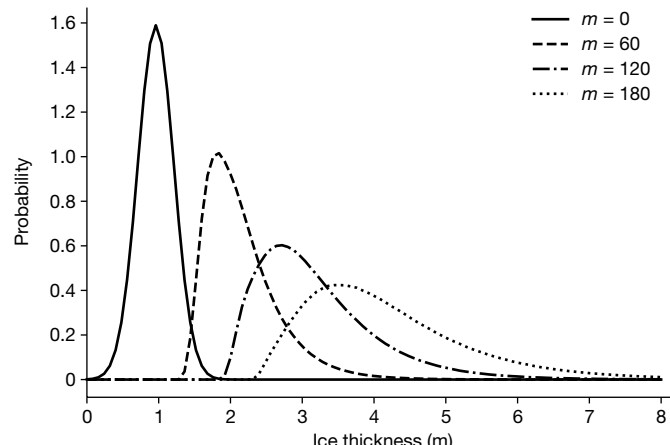

**Fig. 5 | Ice thickness distributions obtained from the stochastic model of dynamic ice thickening.** Probability density functions of $X_m$ for different values of $m$. The smaller $m$ corresponds to a shorter residence time of sea ice in the Arctic Ocean. See the Methods for the descriptions.

in the summer[42,43], which resulted in a perennial increase of ocean heat content in areas of ice formation (Extended Data Fig. 1). After 2007, suppression of winter ice growth due to the accumulated ocean heat became conspicuous[44] and the resultant thinner ice pack became more vulnerable to summer melt in the following year. Prolongation of the summer melt season promoted further ice–albedo feedback[45] and has increased oceanic heat absorption in the summer[7]. Thus, summer ice extent and thickness in areas of ice formation has not recovered to the state before 2007 (Fig. 4c). In addition, continuing weakening of the cold halocline in the Siberian sector also influenced the upper ocean heat content[46] and possibly slowed down ice growth offshore of the Laptev Sea in recent years[17].

Our analysis demonstrates the long-lasting impact of climate change on Arctic sea ice through reduced residence time, suggesting an irreversible response of Arctic sea ice thickness connected to an increase of ocean heat content in areas of ice formation. The large reduction of summer ice extent in the Alaskan and Siberian sectors in 2005 and 2007 triggered intensive ice–albedo feedback[42,45] and initiated the perennial increase of ocean heat content in these areas[44]. This resulted in the stepwise reduction of residence time of sea ice in the Siberian sector of the Arctic, and hence a nonlinear response of the system. Before the shift, sea ice formed in and offshore of the Siberian shelves overwintered (spent about 15 months) in this area before entering the TPD (Extended Data Fig. 2), during which the ice thickened and increased its deformed fraction. After the shift, ice stayed in this area only about 6 months on average (Extended Data Fig. 2), resulting in recruitment of newly formed younger ice into the TPD and more sea ice formation during TPD transit to the Fram Strait[26]. The younger ice is thin, weakly linked and features ridges with more shallow keels; hence, it is more prone to wind forcing pushing the ice towards the Atlantic sector of the Arctic[28,47]. This process accelerated the TPD from 2007 onwards (Fig. 4d), while enhanced wind forcing after 2007 may also have contributed to the acceleration of the TPD (Extended Data Fig. 7). Because of the shorter residence time, the part of the ice that has thermodynamically grown is thinner[17] (reduction of modal thickness in Fig. 1b) and the relative amount of the deformed fraction of ice has decreased (Figs. 1b and 2).

Impacts of this regime shift in Arctic sea ice on the pan-Arctic environment are extensive and require further investigation. Thinner and less deformed sea ice causes reduced momentum exchange between ice and ocean, contributing to reduced mixing in the upper ocean underneath areas that are fully covered with ice. This may affect entrainment of heat and nutrients from subsurface to surface ocean with a potential

consequence on the biogeochemical cycles involving higher trophic levels. By contrast, however, sea ice retreat in marginal ice zones and continuing weakening of the cold halocline in the Atlantic sector allows for more turbulent mixing and winter convection in the upper ocean[46]. These counteracting effects can influence the regional contrasts of the ocean environment between fully ice-covered areas and marginal ice zones in the Arctic. In addition, habitat conditions of younger, level sea ice are different from those in older multi-year ice and might affect the sympagic (ice-associated) communities and their diversity[48,49]. Ridged sea ice supports higher biomass[48] and represents safe havens for organisms to hide from predators[50]. The amount of ridges and deformed ice has also consequences for human activity. Thinner, more level ice is less challenging for ship navigation than in thicker, deformed ice and, along with less ice/shorter ice seasons in general, may allow for an increase in Arctic maritime traffic. Finally, interdisciplinary studies in the Atlantic sector of the Arctic and downstream of the Fram Strait outflow are needed to shed light on the consequences of the described sea ice regime shift and its impacts on physical and biogeochemical processes.

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

## Methods

### Data

Sea ice draft data were obtained from upward looking sonar (ULS) moored in the East Greenland Current in the western Fram Strait. The dataset continuously covers the last three decades (1990–2019) with some short temporal gaps. Four ULSs were zonally aligned at approximately 79° N from 3° W to 6.5° W (Fig. 1a). The latitude of the mooring array changed from 79° N to 78.8° N in 2001. The zonal positions (names) of the moorings equipped with ULS are 3° W (F11), 4° W (F12), 5° W (F13) and 6.5° W (F14), respectively. There are three main temporal data gaps during the three decades of measurements, that is, 1996, 2002 and 2008. Except for these gaps, ULSs were in operation although their number varied from time to time. The ULS measures the travel time of the sound reflected at the bottom of the floating sea ice, from which we calculate the ice draft, the underwater fraction of sea ice[54]. The raw data were processed to ice draft using procedures described in earlier literature[55,56]. The accuracy of each draft measurement ranges from 0.1 m (ice profiling sonars (IPS) deployed after 2006) to 0.2 m (ES300 instruments before 2006), while the uncertainty of each individual measurement is not subject to bias errors and the summary error statistics of monthly values are less than 0.1 m[57].

The daily mean sea ice motion product provided by the National Snow and Ice Data Center (Polar Pathfinder Daily 25 km EASE-Grid Sea Ice Motion Vectors v.4.1, hereafter NSIDCv4 ice drift) was applied for backward trajectory calculation of sea ice floes, estimating residence time of sea ice in the Arctic and analysis of spatial average of ice drift speed. The product was derived from a combination of various ice motion estimates from remote sensing and tracks of ice-tethered buoys[52]. We applied the motion vectors from 1984 to 2019 for the backward trajectory calculation. Sea ice concentration data were taken from the Global Sea Ice Concentration Climate Data Records of the EUMETSAT OSI SAF[51] (OSI-409 v.1.2 for the period from January 1984 to April 2015 and OSI-430 v.1.2 for the period after April 2015). The ice concentration data were used for the sea ice trajectory calculations and to analyse spatial and temporal changes of ice concentration.

Sea surface temperatures obtained from the NOAA/NESDIS/NCEI Daily Optimum Interpolation Sea Surface Temperature v.2.1 (ref. [58]) were used to examine temporal variation in sea surface temperature in areas of ice formation (Extended Data Fig. 1a,b). A dataset of Arctic Ocean in situ hydrographic observations[59] was also used to examine the change in ocean surface temperature between 1990 and 2006 and between 2007 and 2019. The dataset was revised using recent observations[60] and gridded to 110 × 110 km cells covering the whole Arctic Ocean. The mean temperature in each period (1990–2006 and 2007–2019) in each cell was defined by an average of all available measurements for every 3-month period (January–March, April–June, July–September and October–December). If the number of available data in a cell was less than four, a missing value was assigned to the cell. The difference in summer sea surface temperature (July–September, 0–20 m) between two periods (1990–2006 and 2007–2019) was used for the analysis (Extended Data Fig. 1c). Atmospheric data were taken from the European Center for Medium-Range Weather Forecasts Reanalysis v.5 (ref. [61]) (ERA5). Daily and monthly mean sea-level pressure and 10 m wind were used in the analyses.

### Ice thickness distribution

Ice thickness distributions were derived on a monthly basis. All sea ice draft measurements in the Fram Strait from 1990 to 2019 were classified into draft thickness bins of 0.1 m, ranging from 0 to 8 m (80 bins in total). The number of data samples (ice draft measurements) used to derive the distributions varied from time to time. From 1990 to 2005, $O(10^4)$ samples were used to derive the distribution functions on a monthly basis (measurement interval of 240 s in most cases);

after 2006, $O(10^6)$ samples were used (interval of 2 s). The number of samples used were sufficient to derive thickness distribution on a monthly basis[57]. The number of data in each bin were divided by the total number of measurements to derive the distribution function. The open water fraction (that is, zero ice thickness) was excluded when deriving the function. Distribution functions, including the open water fraction, are shown in Extended Data Fig. 3. If the temporal coverage of the data samples was less than 15% of a monthly coverage, the distribution function was not defined and removed from the analysis. The draft thickness distributions were converted to ice thickness distributions by an average ratio of draft to thickness in the Fram Strait, 1.136 (ref. [62]). Although the ratio has some seasonal variability and might have slightly changed due to changes in ice density and snow load in different seasons and years, we assumed that the change was not considerable for the aim of the current analyses. A composite time series of ice thickness distribution (Fig. 2a and Extended Data Fig. 3) was obtained from a combination of available distribution functions from F11 to F14. More specifically, one of the distribution functions from F11 to F14 was used with a priority order of F13, F14, F12, F11. The composite time series is shown in Fig. 2a, while the time series at each site are shown in Extended Data Fig. 4. The fractions of sea ice thicker than two thresholds, 4 and 5 m, were calculated as the cumulative function of all available distributions (that is, all distributions from F11 to F14) in each month and are shown in Fig. 2d.

The uncertainties on the estimates of monthly fractions of ice thicker than a threshold and position of the modal ice peak were assessed numerically using a moving block bootstrap approach[63]. Bootstrapping is a family of resampling techniques used to derive uncertainties on various complex estimators for large datasets and uses random sampling with replacement[64]. The presence of autocorrelation in ice draft series from IPS (ULSs deployed after 2006) suggested using the moving block bootstrap approach[63]. The method splits the original monthly series of ice drafts $O(10^6)$ samples of length $N$ into $N − K + 1$ overlapping blocks of length $K$ each. The block length was set to 30 samples that approximately corresponded to a distance of 10 m covered by ice travelling at 0.3 knots. It roughly corresponded to the lower limit of a horizontal spatial scale of ice ridges. For ES300 instruments (ULSs deployed until 2005) with a lower sampling rate of 240 s, ordinary bootstrapping was used.

At each of $M$ steps of bootstrap sampling, $N/K$ blocks were drawn at random, with replacement, from the constructed set of $N − K + 1$ blocks, making a new bootstrap data sample for the month. A Gaussian noise of $N(0,1)$ was further added to the data to account for measurement uncertainty. The mean and s.d. of the fractions of thicker ice and modal ice peak position were then calculated directly from the $M$ estimates derived at each step of the procedure. The results suggested that the monthly coefficient of variation or a ratio of the s.d. of the estimate to its mean, for the IPS data varies from 1% to 3% for both fractions, being on average lower (1–2%) for a fraction of ice greater than 4 m thick. For ES300, the coefficient of variation was slightly higher at about 4(6)% for the fractions of ice thicker than 4(5) m. The same applied to the position of the modal peak, which showed a coefficient of variation of 0–3% being typically closer to 0 for the IPS data. It suggested that the selected bin width was large enough to accommodate uncertainties related to the approach and data. For the ES300 data, the monthly s.d. of the modal peak position was higher, up to 30 cm, and the coefficient of variation was within 9%. Therefore, we postulate that the inferred uncertainties are far too low to have any noticeable influence on the results of the shift detection analysis.

### Modal peak and variance of ice thickness distributions

As statistical measures of ice thickness distributions, we examined modal thickness and variance. The monthly mean ice thickness distributions (740 samples in total) were fitted to log-normal functions:

$$F(x) = \frac{1}{\sqrt{2\pi\sigma_f^2}\,x}\exp\left(\frac{-(\ln x - \mu_f)^2}{2\sigma_f^2}\right) \tag{4}$$

where $\sigma_f$ and $\mu_f$ are the fitting parameters, $x$ is the ice thickness bin and $F$ is the distribution function. To detect the second peak of the distributions that represents multi-year ice travelled across the Arctic Basin, a cut-off threshold was introduced. The threshold was used to exclude thin sea ice fraction, which is supposedly formed in the vicinity of the Fram Strait and is not representing basin-wide changes of ice properties in the Arctic. We defined the threshold by the minimum between the first and second peak of each monthly mean distribution. A set of two consecutive negative gradients (towards thicker bins) followed by two consecutive positive gradients was used to detect the minimum (after applying 3-bin smoothing), while a threshold of 1.53 m (corresponding to 1.3 m of ice draft) was applied when an estimated threshold was thicker than 3 m. Function values ranging lower than the threshold were set to zero (zero case) or excluded from the fitting (NaN case). A least-square minimization was applied to fit a log-normal function to the distribution. In general, the fitted log-normal functions represent the distribution very well. The NaN case slightly underestimates the modal peak, while the zero case captures the peak very well. Examples of distribution functions, together with cut-off thresholds and fitted log-normal functions, are shown in Extended Data Fig. 5. The modal peak of the log-normal function roughly gives the thickness of thermodynamically grown sea ice, while the variance of the function quantifies the deformed fraction of sea ice (dynamically thickened thickness). Changes in modal peak height and variance of the fitted log-normal functions, $\mathrm{var}(x) = \exp(2\mu_f + \sigma_f^2)(\exp(\sigma_f^2) - 1)$, for the last three decades are summarized in Fig. 2b,c. The time series of modal thickness and the fitting parameters $\sigma_f$ and $\mu_f$ are summarized in Extended Data Fig. 6.

### Regime shifts detection
A sequential algorithm for regime shift detection[23] was applied to all time series. The method identifies discontinuities in a time series using a data-driven approach that does not require an a priori assumption on the timing of the regime shifts. The method first identifies potential change points sequentially by checking if the anomaly of the data point is statistically significant from the mean value of the current regime. If it is significant, the following data points are sequentially used to assess the confidence of the shift, using a regime shift index (RSI). RSI represents a cumulative sum of normalized deviations from the hypothetical mean level for the new regime, for which the difference from the mean level of the current regime is statistically significant according to a Student's $t$-test. If the RSI is positive for all points sequentially within the specified cut-off length, the null hypothesis of a constant mean is rejected. This led us to conclude that the regime shift might have occurred at that point in time[65]. If multiple data are available at a certain point in time (that is, multiple sites from F11 to F14), the mean value is applied in the time series. Before testing, the temporal gaps of the time series were interpolated by the average of all available data (modal peak height (Fig. 2b), variance (Fig. 2c) and modal thickness (Extended Data Fig. 6a) of ice thickness distributions, fraction of thick sea ice (Fig. 2d) and residence time of sea ice (Fig. 3a and Extended Data Fig. 2)). The cut-off length was set to 7 years (84 months) to cover the advection timescale (travelling time across the Arctic) of sea ice, while at the same time, detecting shifts occurring at a timescale shorter than a decade. Other cut-off lengths (3, 4, 5, 6, 8 and 10 years) were also tested to see the sensitivity and robustness of the results. A summary of the test results is given in Extended Data Table 1; the timing of the detected shifts is shown in all time series except for Fig. 2. The timing of the detected shifts of modal peak height and variance of ice thickness distributions are shown in Extended Data Fig. 6b,c, while those of the fraction of thick sea ice are shown in Extended Data Fig. 8.

RSIs, respective $P$ values and the shift of the means are summarized in Extended Data Table 2.

### Sea ice trajectory analysis
To investigate changes of pathways and residence time of sea ice in the Arctic basins, sea ice trajectories were calculated for the last three decades. Eight pseudo-ice floes were settled in the western half of the Fram Strait section (from prime meridian to 10° W) at the same time and advected backwards in time. The calculations started on the 15th of every month from 1990 to 2019. Daily sea ice motion vectors from the NSIDCv4 were used to update daily position of ice floes backwards in time. Ice motion vectors at the respective floe positions were calculated by interpolation of surrounding points with Gaussian-type weighting (e-folding scale of 25 km). Each trajectory calculation was performed 6 years back in time, while it was terminated if no motion vector was available within a 25-km distance or sea ice concentration at the floe position was lower than 15%. The sea ice concentration at the ice floe position was obtained from OSI-409/OSI-430 with a Gaussian-type weighting (e-folding scale of 12.5 km). The position of each trajectory termination was used to define the location of 'initial sea ice formation'. Trajectories shorter than three months were excluded from the analysis because they represent ice floes formed in the vicinity of the Fram Strait.

Uncertainty of the daily position of the pseudo-floes was assessed by comparisons with ice-tethered buoy tracks obtained from the International Arctic Buoy Program (IABP)[53]. We used 83 buoy tracks that arrived in the Fram Strait from 2000 to 2018 and calculated the corresponding pseudo-buoy tracks backwards in time. The comparisons showed that the mean error of the daily pseudo-buoy positions can be reasonably approximated by a linear function of backtracking days[19], error = 50 + (backtracking days)/2 km. We applied this empirical formula as an error of the daily position of the backward trajectories from 0 to 500 backtracking days, which corresponds to a 200 (300) km error after 300 (500) backtracking days. Note that this error estimate may underestimate the uncertainty because IABP buoy tracks have been included in the NSIDCv4 ice motion product. However, comparisons between non-IABP buoys and pseudo-buoy tracks derived from the NSIDCv4 with error estimates by a bootstrap method showed that pseudo-tracks are largely parallel to the corresponding buoys and the error does not monotonically increase over time[66]. The estimated error circles (approximately 300 km) of ice formation location in the present study are sufficiently small compared to the polygons in Fig. 3b (greater than 1,500-km width), which guarantees the robustness of the analysis.

The residence time of sea ice in the Arctic basins was defined by the period from the start to the termination date of each trajectory. We calculated an average of residence time of eight pseudo-ice floes that arrived in the Fram Strait at the same time and used it to define the mean residence time of ice floes for each month. The uncertainty of the residence time was defined by the s.d. of the residence time of the eight pseudo-ice floes.

### Stochastic model of dynamic ice thickening
The log-normal form of the ice thickness distribution can be obtained from a simple proportionate growth process, $X_m = X_0\Pi_{i=0}^{i=m-1}a_i$. If $\ln X_0$ and $\ln a_0, \ln a_1, .., \ln a_i$ are uncorrelated; thus, the probability function of $X_m$ for large $m$ is given by:

$$f(X_m) = \frac{1}{\sqrt{2\pi\acute{\sigma}(m)^2}\,X_m}\exp\left(\frac{-(\ln(X_m/\acute{X}(m)))^2}{2\acute{\sigma}(m)^2}\right) \tag{5}$$

where $\acute{X}(m) = e^{vm}$ and $\acute{\sigma}(m) = \sigma m^{1/2}$ and $v$ and $\sigma^2$ are the mean and variance of the population distribution of $\ln a_m$ (including $X_0$), respectively[67].

In this study, we provide a concept and description of a stochastic model that formulates sea ice thickening associated with dynamic ice deformation. The model formulates three features of dynamic sea ice thickening by ridging and/or rafting: (1) dynamic ice thickening is a stochastic process (areal and thickening stochasticity); (2) thicker ice has a larger potential to get thicker than thinner ice at a dynamic event (proportionate ice thickening); and (3) thinner ice has a higher probability of dynamic deformation due to its weaker ice strength (preferential deformation of thinner over thicker ice types). The first point consists of two types of stochasticity in the dynamic ice thickening process. One is areal stochasticity, corresponding to the fact that ice deformation only occurs for a small fraction of the pack ice while the rest of the ice is unchanged when a dynamic event occurs. The other is thickening stochasticity, representing the fact that the thickness gain by ridging/rafting varies in space and differs between events. The second point represents a sea ice characteristic that thicker ice is tolerant and can exert stronger compressive force on the ice forming ridges and/or rafts; hence, more energy is potentially available for the dynamic thickening[30,31]. The third point represents the fact that the thinner part of the pack ice is preferentially ridged/rafted when a dynamic event occurs[68,69]. This also takes into account the effect of ice thickness changes on the dynamic thickening process, for example, the thinner ice condition in recent years increases the likelihood of ice deformation.

The two first points can be formulated by a proportionate thickening of stochastic ice thickness:

$$X_i = a_{i-1}X_{i-1} \tag{6}$$

where $X$ is the stochastic ice thickness at a certain location, $i$ denotes the time index counting sporadic dynamic events (for example, passage of a storm) and $a_{i-1}$ is the conditional proportionate thickening increment due to the event at $i-1$. The increment $a_{i-1}$ is a stochastic variable representing both areal and thickening stochasticity. They are implemented as:

$$a_i = \begin{cases} 1 + br_i & \text{with } \alpha_i(\%) \text{ probability} \\ 1 & \text{with } 1 - \alpha_i(\%) \text{ probability} \end{cases} \tag{7}$$

where $b$ is a proportionate thickening constant, $r_i$ is a stochastic thickening increment that represents the thickening stochasticity of $i$-th dynamic event and $\alpha_i$ is the areal probability of dynamic thickening that represents areal stochasticity and gives the probability of ice thickening occurrence. This formula indicates that when a dynamic event occurs, $\alpha(\%)$ area of pack ice experiences dynamic thickening (ridging/rafting), while the rest $(1 - \alpha(\%))$ is unchanged (areal stochasticity). The thickness gain, $br_i$, in the dynamic thickening area, is also a stochastic variable: the possible maximum gain is $b$ while the minimum is 0 ($r_i$ is random, so $0 \leq r_i < 1$) (thickening stochasticity).

We applied the proportionate thickening constant as $b = 0.4$. The choice of $b = 0.4$ implies that sea ice in the ridging/rafting area gains 0.4 m thickness at maximum (0.2 m on average) when the ice is initially 1-m thick, while it gains a 1.2 m thickness at maximum (0.6 m on average) when initially the ice is 3-m thick. The value of the parameter $b$ comes from a recent high-resolution observation ($5 \times 5$ m resolution covering a 9-km² area) of single ice deformation event north of Svalbard, which describes changes in the sea ice freeboard just before and right after a storm event[70]. According to this study, the change in sea ice freeboard in a converging area is 0.07 m (from 0.36 m to 0.43 m) on average, corresponding to 0.58 m dynamic thickening of ice by ridging and/or rafting (assuming the freeboard to thickness ratio = 8.35)[62]. The gain relative to the mean ice thickness is estimated by $b = 0.58/1.45 = 0.4$ (the mean ice thickness in the survey area = 1.45 m). We applied the value for the proportionate thickening constant, $b$, as the first approach to develop the model. Although the study captured detailed spatial change in the sea ice freeboard, the estimate of $b$ comes from a one-time

event and hence needs further assessments by future observations. It should be noted that $b = 0.4$ is an areal average estimated from the observations, whereas we applied $b$ as the upper bound of the proportionate thickening. This is because the probability function of the stochastic thickening increment, $r_i$, is not known so far; hence, we assumed a constant probability of the increment between 0 and $b$, potentially causing excessive thickening near the upper bound. The effect of the choice of $b$ is discussed below.

The areal probability of dynamic thickening, $\alpha$, is included to take the third point into account, that is, thinner ice has more chance to be ridged and/or rafted than thicker ice when a dynamic event occurs. To implement this feature, $\alpha$ is given by a function of ice thickness: the areal probability is inversely proportional to the stochastic ice thickness $X_i$:

$$\alpha_i(X_i) = \frac{8}{X_i + 1}(\%). \tag{8}$$

The formula indicates that 1-m thick ice experiences dynamic thickening at 4% areal probability, while 3-m ice experiences dynamic thickening at 2% areal probability. Our first implementation of this formula is based on an observational estimate of areal fraction of dynamic thickening[70]. According to the high-resolution survey of a single dynamic event, thickening occurred in 4% of the survey area with a mean ice thickness of 1.45 m. This formula also needs further evaluation by comparing with future observations that address the relationship between areal probability of deformation and ice thickness.

Another parameter necessary for the model is the number of dynamic events, $m$, that is, external forcing that could cause mechanical fracturing of sea ice and consequent ridging and/or rafting. We used the number of Arctic cyclones that passes over the ice pack as a first-order indicator of the number of dynamic events. Typically 90–130 cyclones per year occur in the Arctic Ocean (40–60 cyclones in winter, 50–70 cyclones in summer)[71]. A typical size of an Arctic cyclone is approximately $3 \times 10^6$ km² (mean radius of approximately $10^3$ km)[71], which covers approximately one third of the ice-covered area of the Arctic Ocean. We therefore assumed that one-third of all cyclones hits the ice pack at a certain location in the Arctic, that is, the ice pack experiences approximately 40 dynamic events per year. This corresponds to approximately 80–240 dynamic deformation events for the typical residence time of sea ice in the Arctic (2–6 years; Fig. 3a).

In addition, examples shown in Fig. 5 contain a simple thermodynamic term to mimic the effect of modal peak shift of thickness distribution due to thermodynamic ice growth:

$$X_i = a_{i-1}X_{i-1} + c/X_{i-1} \tag{9}$$

where $c$ is the thermodynamic ice growth coefficient. This term comes from a simplified thermodynamic process without thermal inertia of sea ice and heat flux from the ocean[72]:

$$\frac{dH}{dt} = \frac{\kappa_{\text{ice}}(T_f - T_s)}{\rho_{\text{ice}}L_f H} \tag{10}$$

where $H$ is the ice thickness, $\kappa_{\text{ice}}$ is the heat conductivity of ice, $\rho_{\text{ice}}$ is the ice density, $L_f$ is the latent heat of freezing, $T_f$ is the freezing temperature of sea ice and $T_s$ is the temperature at the ice surface. We applied this formula with a simplification, $\Delta H = c/H$, where $\Delta H$ is the ice thickness change due to a thermodynamic process, $c$ is a thermodynamic ice growth coefficient corresponding to $\Delta t \kappa_{\text{ice}}(T_f - T_s)/(\rho_{\text{ice}}L_f)$. As the model does not include a process that forms new thin ice by lead opening, inclusion of the thermal forcing term without a compensating term makes the modal peak very steep after few years, that is, no ice exists in thickness ranges thinner than the thermal equilibrium thickness. To alleviate such an excessive modal peak generation and to take into account the insulating effect of the snow pack that substantially delays

thermodynamic ice growth, we applied a moderate value, $c = 0.015$, which is about one-third of the value estimated from $c = \Delta t \kappa_{ice} (T_f - T_s)/(\rho_{ice} L_f)$, where $\Delta t \cong 9$ d (corresponding to 40 dynamic events per year) and annual mean surface air temperature of $T_s = 263$ K.

The initial condition of the ice thickness distribution in Fig. 5 is given by a thermodynamically grown sea ice without dynamic deformation, $X_0$. This is also a stochastic variable, having a normal distribution for simplicity:

$$g(X_0) = \frac{1}{\sqrt{2\pi}\,\sigma_0} \exp\left(\frac{-(x-\mu_0)^2}{2\sigma_0^2}\right) \tag{11}$$

where $\mu_0 = 1.0$ and $\sigma_0 = 0.25$ are applied in the examples (that is, 1 m mean ice thickness with 0.25 s.d., shown by $m = 0$ in Fig. 5), which roughly corresponds to the thickness of new ice three months after its formation (based on Anderson's freezing degree days law[25], with an assumption of $T_s = 253$ K). Figure 5 shows examples of ice thickness distribution after 60, 120 and 180 dynamic events, roughly corresponding to 1.5, 3 and 4.5 years of residence time of sea ice.

The current formulation contains three parameters to describe the dynamic ice thickening process: $b$ (the proportionate thickening constant); $\alpha$ (the areal probability of dynamic thickening); and $m$ (the number of dynamic events). In this study, we briefly describe the sensitivity of the ice thickness distributions to these parameters. In general, a smaller (larger) thickening constant $b$ decelerates (accelerates) the dynamic thickening process, that is, a smaller $b$ gives a smaller variance and steeper modal peak of thickness distribution if $\alpha$ and $m$ are fixed. However, a large value of $b$ (for example, $b = 0.8$, indicating that ridged ice can be 1.8 times thicker than the ice before an event at maximum) makes the distribution bimodal because the possible thickness gain at each dynamic event is far from the modal thickness and the ridged/rafted ice tends to generate another peak apart from the mode. Therefore, the possible and realistic range of $b$ should be examined further together with the probability density function of the thickening increment $r$ by high-resolution observations in the future. The areal probability of dynamic thickening, $\alpha$, also affects the evolution of the dynamic thickening process. A larger $\alpha$ promotes dynamic thickening because a larger fraction of pack ice can be deformed at one event. The thickness dependency of the probability, equation (8), decelerates further thickening of thick ice. Although values of $b$ and $\alpha$ affect the progress of dynamic thickening in the model, we obtained similar ice thickness distributions with a log-normal form sooner or later, that is, smaller $b$ and $\alpha$ can be compensated by a large $m$, the number of deformation events, indicating a robustness of the formulation. The resulting shape of the distribution, its temporal evolution (Fig. 5) and its comparison with the observed change in distribution (Fig. 1b) suggest that the proposed stochastic ice thickening model captures the essence of the dynamic thickening process that resulted in the observed changes in ice thickness distribution.

## Data availability

The ice thickness distribution data from the Fram Strait Arctic Outflow Observatory are available from the Norwegian Polar Data Centre, https://data.npolar.no/dataset/b94cb848-3120-4f29-a827-298108e0d059. The ice drift data are available at the National Snow and Ice Data Center https://nsidc.org/data/NSIDC-0116/versions/4 (NSIDCv4). The sea ice concentration data used are available at ftp://osisaf.met.no/reprocessed/ice/conc/v2p0/ (OSI-409, superseded by OSI-450) and

ftp://osisaf.met.no/reprocessed/ice/conc-cont-reproc/v1p2/ (OSI-430). The ERA5 reanalysis product is available at https://cds.climate.copernicus.eu/#!/home.

## Code availability

The backward trajectory code is available at *Zenodo* https://zenodo.org/record/7390660#.Y4oSL9LMIUF (https://doi.org/10.5281/zenodo.7390659).

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

**Acknowledgements** This study has been made possible by the long-term observations from the Fram Strait Arctic Outflow Observatory maintained by the Norwegian Polar Institute. This work was supported by the Norwegian Research Council (grant no. 286971, project FreshArc) and partly received funding from the European Union's Horizon 2020 research and innovation programme under grant no. 101003826 via project CRiceS (Climate Relevant interactions and feedbacks: the key role of sea ice and Snow in the polar and global climate system). We thank J. Haapala and M. Lensu for discussions about the proportionate ice growth model.

**Author contributions** H.S. and D.V.D. conceptualized the study. H.S., D.V.D. and M.A.G. devised the methodology. H.S. processed the data, devised the theoretical model, carried out the formal analysis and data visualization and wrote the original draft. H.S. and D.V.D. carried out the uncertainty assessments. H.S., L.d.S., D.V.D., S.G. and M.A.G. carried out the investigation. H.S., L.d.S., S.G., D.V.D. and M.A.G. reviewed and edited the manuscript. L.d.S. (FreshArc) and M.A.G. (CRiceS) acquired the funding. L.d.S. was the project lead and organized the study.

**Competing interests** The authors declare no competing interests.

**Additional information**
**Correspondence and requests for materials** should be addressed to Hiroshi Sumata.

**a**

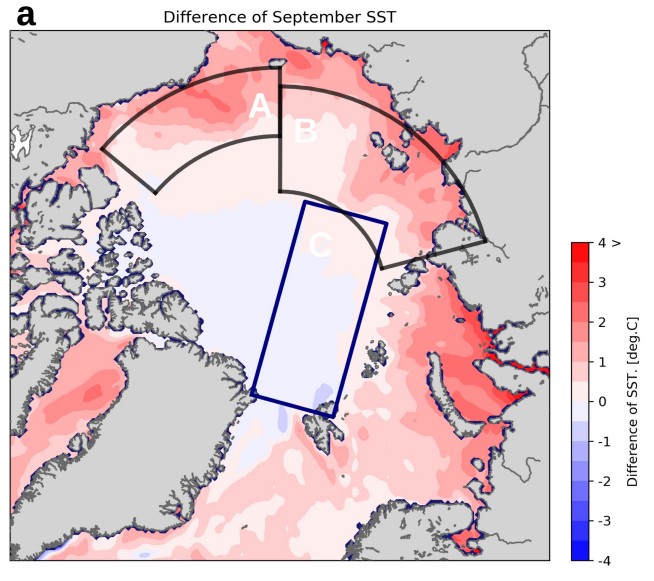

Difference of September SST

**b**

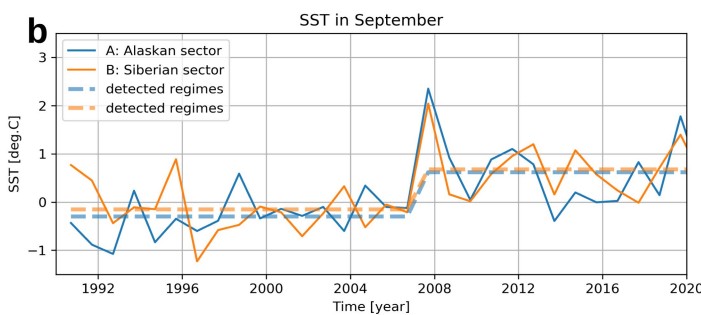

SST in September

**c**

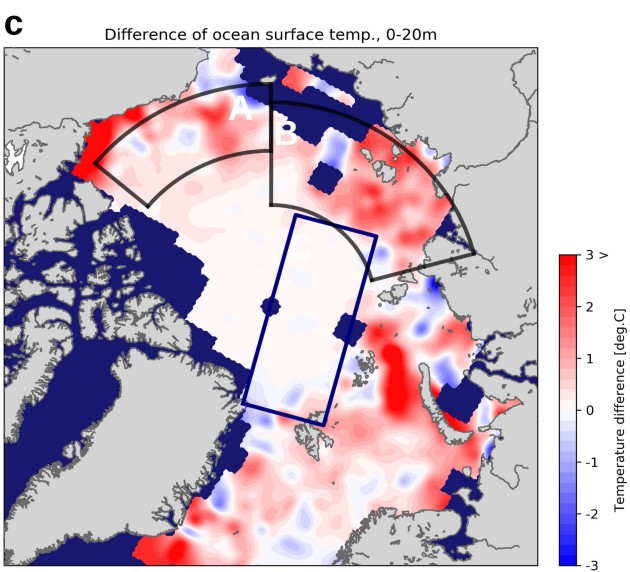

Difference of ocean surface temp., 0-20m

**Extended Data Fig. 1 | Difference of sea surface temperature (SST) between two periods 1990–2006 and 2007–2019.** (a) Difference of mean September SST estimated from Daily Optimum Interpolated Sea Surface Temperature data set (DOISST ver. 2.1)[58], (b) Time series of mean September SST in sea ice formation areas A and B, calculated from DOISST. (c) Difference of upper ocean temperature (July to September mean, 0–20 m) between the two periods calculated from in-situ observational datasets[59,60]. The dashed lines in (b) denote detected regimes by sequential t-test described in Methods. Matplotlib basemap toolkit is used to plot the map.

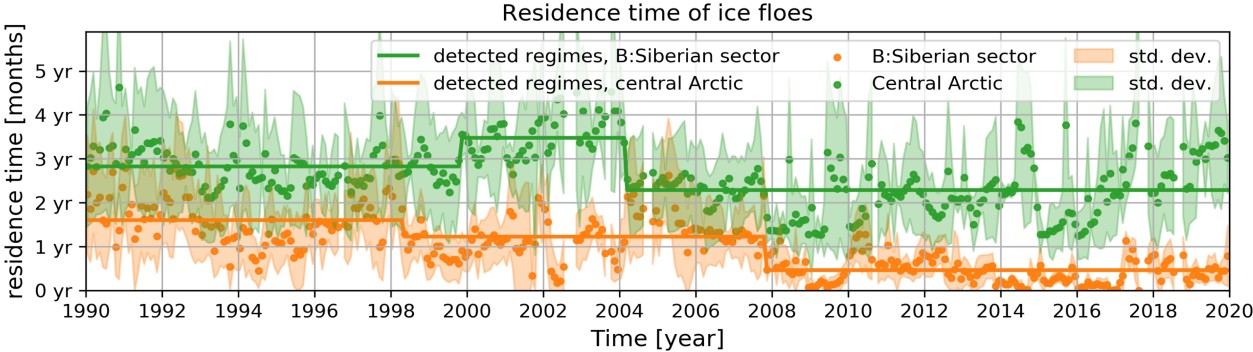

**Extended Data Fig. 2 | Mean residence time of sea ice in the Siberian sector and the central Arctic.** The residence time is calculated by the backward trajectories described in Methods. The central Arctic is defined being outside of the two polygons A and B. The ice formation areas A and B are shown in Fig. 3b in the main text. The solid lines denote regimes detected by the sequential t-test described in Methods.

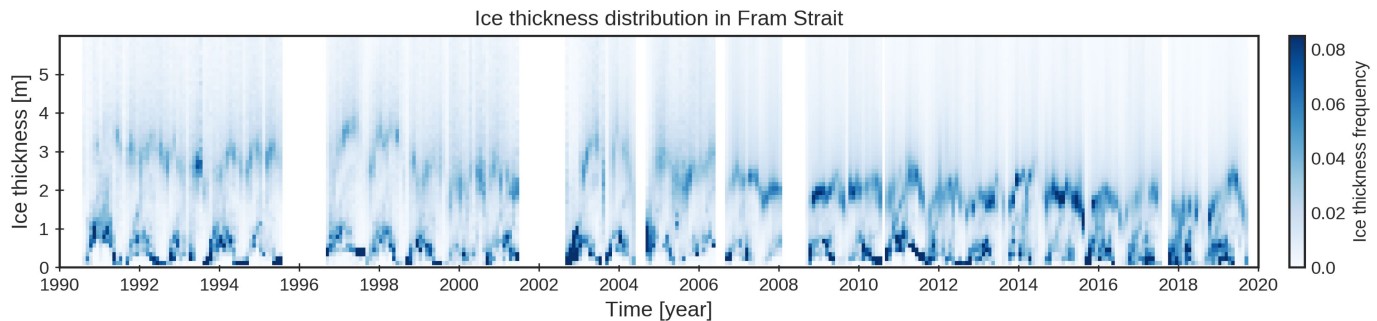

**Extended Data Fig. 3 | Time series of ice thickness distribution in Fram Strait including open water fraction.** The thickness distributions are derived including open water fraction (i.e., zero thickness bin) on monthly basis. Data processing procedures are described in Methods.

Time series of ice thickness PDF

threshold of temporal coverage to define PDF = 0.15
conversion factor from draft to thickness = 1.136
binning range: 0.0 to 8.0 [m]
bins width = 0.1 [m]

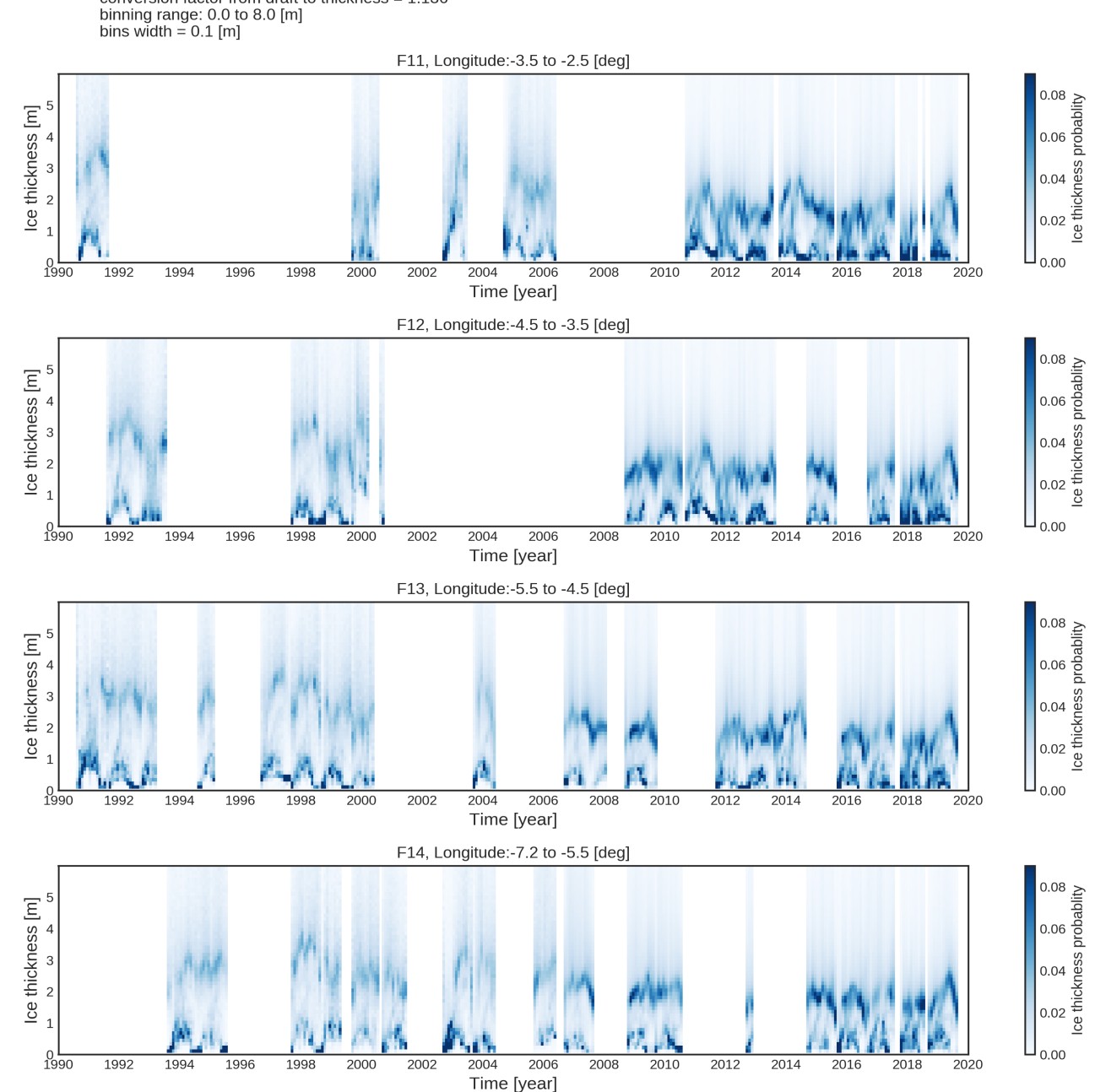

**Extended Data Fig. 4 | Time series of ice thickness distribution in each site.** Time series of ice thickness distribution observed by each moored ULS (F11 to F14) in Fram Strait. The distributions are derived on monthly basis. Data processing procedures are described in Methods.

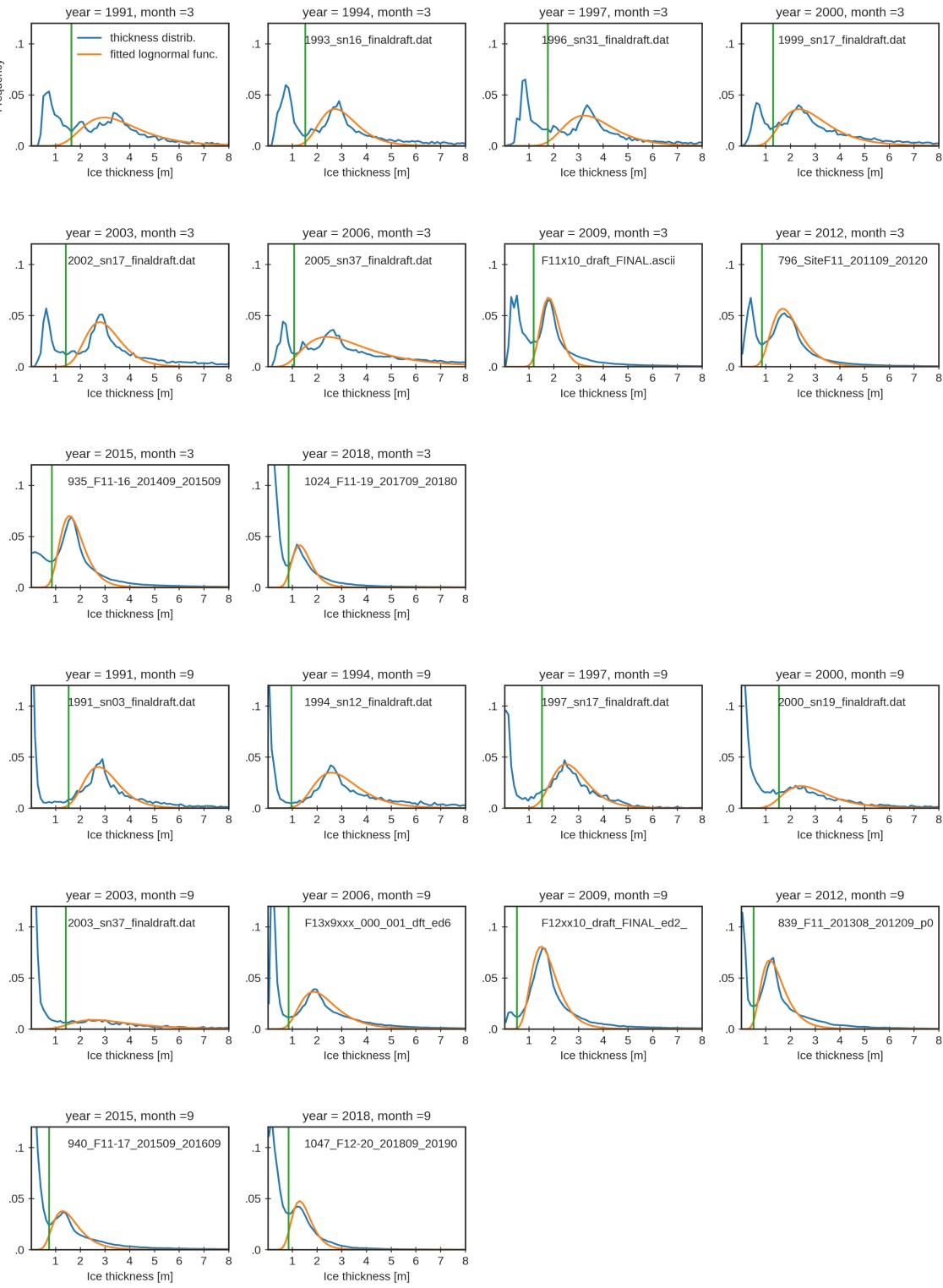

**Extended Data Fig. 5 | Examples of winter and summer sea ice thickness distributions in Fram Strait.** Examples of sea ice thickness distributions and corresponding fitted lognormal functions in March (top three rows) and September (bottom three rows). The blue, orange and green lines show ice thickness distribution, fitted lognormal function, and cut-off threshold, respectively. The plots are shown for every three years if data are available. Data processing procedures are described in Methods.

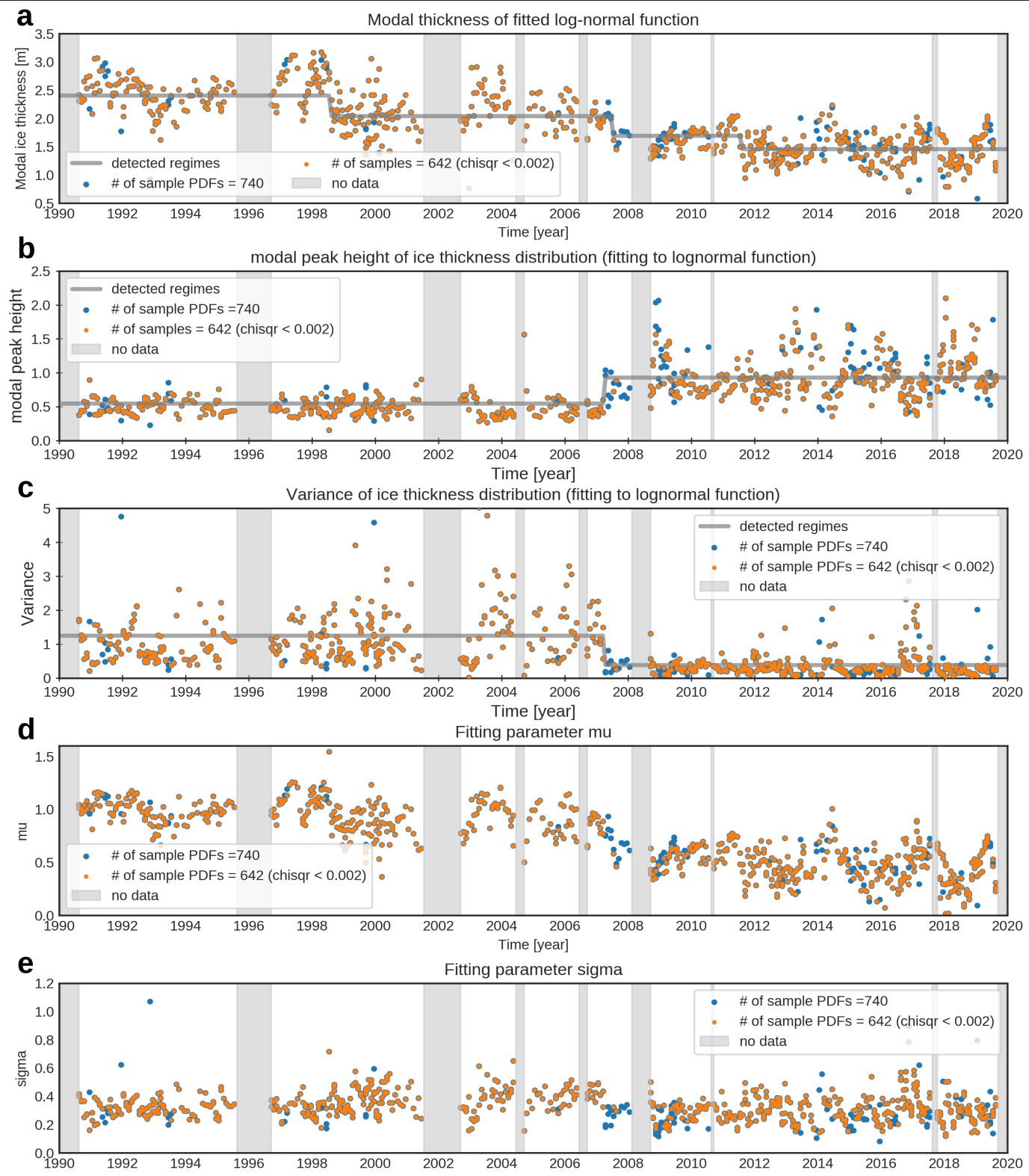

**Extended Data Fig. 6 | Time series of modal peak height and fitting parameters.** Time series of (a) modal thickness, (b) modal peak height, (c) variance, and (d, e) fitting parameters of lognormal functions. Data processing procedures are described in Methods. The gray solid lines in panels (a–c) show regimes detected by the sequential t-test.

**a** Difference of SLP and wind in summer

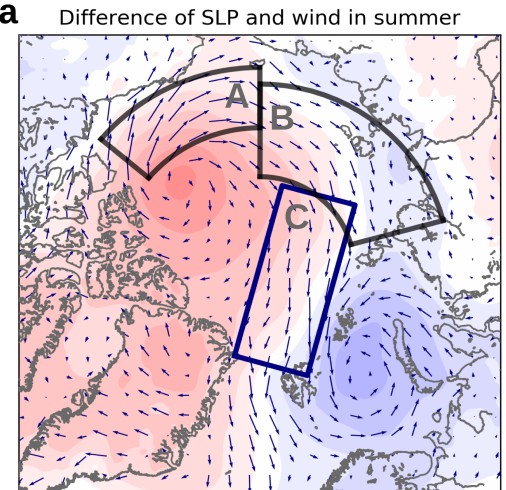

**b** Difference of SLP and wind in winter

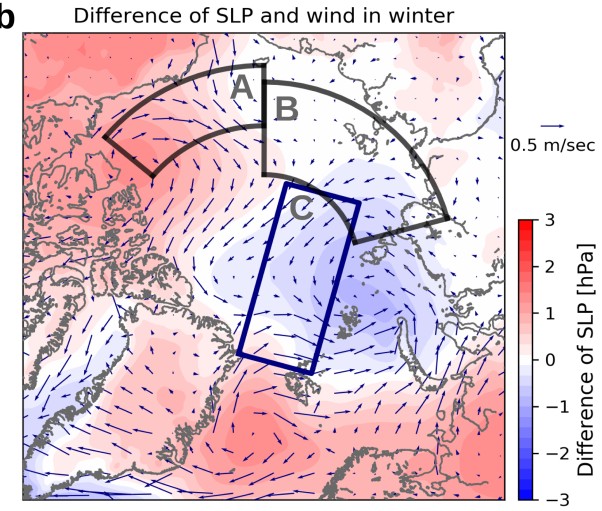

**c**

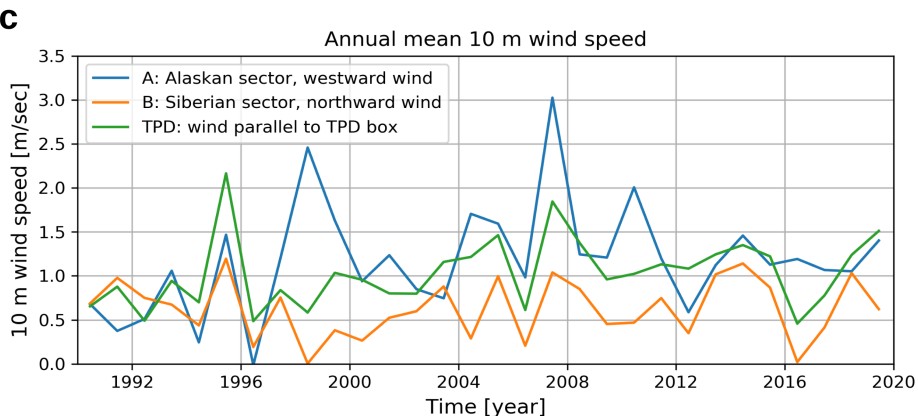

**Extended Data Fig. 7 | Change of sea level pressure and wind pattern after the regime shift.** Difference of sea level pressure (SLP) and wind field between two periods 1990–2006 and 2007–2019, in (a) summer (from June to November) and (b) winter (from December to May), and (c) time series of annual mean 10 m wind averaged in the three polygons shown in panels (a) and (b). The polygons show (a) Alaskan sector, (b) Siberian sector, and (c) the area representing the Transpolar Drift Stream. The mean speed in the rectangular box C is the component of 10 m wind vector parallel to the major axis of the box (positive wind speed orients to the Fram Strait). SLP and 10 m wind data are taken from ERA5[61]. Matplotlib basemap toolkit is used to plot the map.

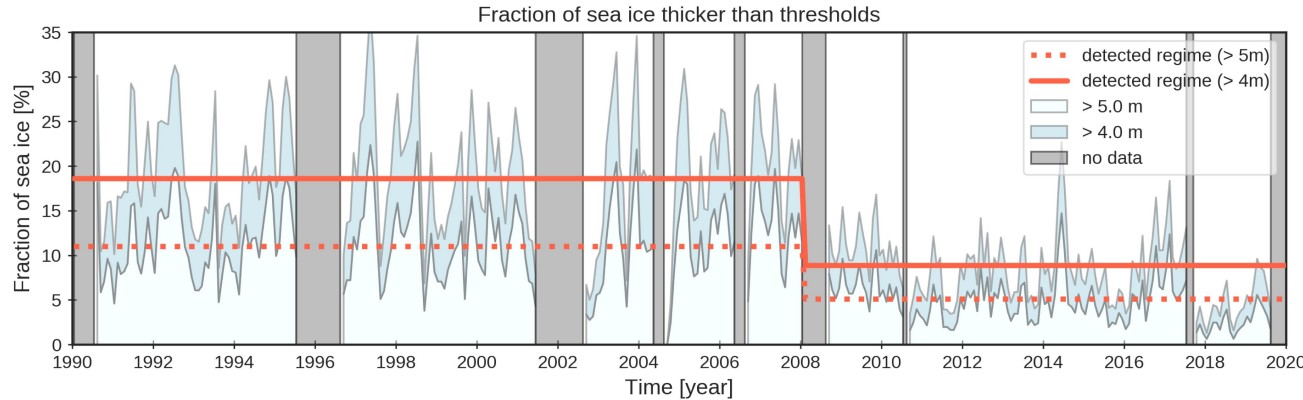

**Extended Data Fig. 8 | Fraction of thick sea ice with detected regimes.** Fraction of thick sea ice (> 5 m and > 4m) observed in Fram Strait. Data processing procedures are described in Methods. The solid and dashed lines denote regimes detected by the sequential *t*-test.

**Extended Data Table 1 | Summary of sequential *t*-test analysis of the regime shift detection**

| Province | Variable | Timing of the detected regime shift (cut-off length: 7 years) | Corresponding figure & notes |
|---|---|---|---|
| Fram Strait | Variance of ice thickness PDFs | Apr. 2007 | Fig. 2c, ED Fig. 6c |
| | Modal peak height of ice thickness PDFs | Apr. 2007 | Fig. 2b, ED Fig. 6b |
| | Fraction of thick ice (> 5 m) | Feb. 2008 | Fig. 2d, ED Fig. 8 |
| | Fraction of thick ice (> 4 m) | Feb. 2008 | Fig. 2d, ED Fig. 8 |
| | Modal ice thickness | Aug. 1998, Jul. 2007, Aug. 2011 | ED Fig. 6a |
| Alaskan sector | September sea ice concentration | 1998, 2007 | Fig. 4c |
| | Westward ice drift speed | 2007 | Fig. 4d |
| | September sea surface temperature | 2007 | ED Fig. 1b |
| Siberian sector | September sea ice concentration | 2005* | Fig. 4c, *shift is detected in 2007 instead of 2005, if cut-off length of 3 or 4 years is applied. |
| | Northward sea ice drift speed | No shift detected** | Fig. 4d, **shift is detected in 2007 if cut-off lengths longer than 8 years are applied (shown in Fig. 4d). |
| | September sea surface temperature | 2007 | ED Fig. 1b |
| | Residence time of sea ice | May 1998, Nov. 2007 | ED Fig. 2 |
| TransPolar Drift | Sea ice drift speed | 2007 | Fig. 4d |
| | Residence time of sea ice | Nov. 1999***, Mar. 2004 | ED Fig. 2, *** only the 2004 shift is detected if cut-off lengths longer than 8 years are applied. |
| Pan-Arctic | Residence time of sea ice | Jan. 1993, Sep. 2005, Nov. 2007 | Fig. 3a |

**Extended Data Table 2 | Details of sequential *t*-test analysis of the regime shift detection**

| Province | Variable [unit] | Timing of the detected regime shift (regime shift index) | *p*-value | Shift of the mean | Note |
|---|---|---|---|---|---|
| Fram Strait | Variance of ice thickness PDFs [no dim.] | 2007 (0.45) | 6.6e-13 | -0.87 | |
| | Modal peak height of ice thickness PDFs [no dim.] | 2007 (1.15) | 1.5e-37 | +0.38 | |
| | Fraction of thick ice (> 5 m) [%] | 2008 (1.09) | 3.4e-45 | -5.9 | |
| | Fraction of thick ice (> 4 m) [%] | 2008 (1.19) | 3.2e-50 | -9.7 | |
| | Modal ice thickness [m] | 1998 (0.76)<br>2007 (1.10)<br>2011 (0.17) | 1.6e-14<br>5.4e-19<br>2.2e-10 | -0.36<br>-0.35<br>-0.23 | |
| Alaskan sector | September sea ice concentration [%] | 1998 (0.68)<br>2007 (0.40) | 2.6e-4<br>1.2e-4 | -21<br>-20 | |
| | Westward ice drift speed [cm/sec] | 2007 (0.29) | 6.4e-5 | +1.4 | |
| | September sea surface temperature [°C] | 2007 (0.49) | 4.5e-4 | +0.95 | |
| Siberian sector | September sea ice concentration [%] | 2005 (1.14) | 1.8e-8 | -30 | |
| | Northward sea ice drift speed [cm/sec] | 2007* (0.32) | 4.4e-3 | +0.49 | *the shift is detected with cut-off length of 8 years |
| | September sea surface temperature [°C] | 2007 (0.21) | 3.4e-4 | +0.84 | |
| | Residence time of sea ice [year] | 1998 (0.48)<br>2007 (1.14) | 2.4e-7<br>2.8e-33 | -0.38<br>-0.76 | |
| TransPolar Drift | Sea ice drift speed [cm/sec] | 2007 (0.47) | 9.6e-4 | +0.86 | |
| | Residence time of sea ice [year] | 1999 (0.08)<br>2004 (1.16) | 6.8e-12<br>3.0e-25 | +0.65<br>-1.19 | |
| Pan-Arctic | Residence time of sea ice [year] | 1993 (0.19)<br>2005 (1.46)<br>2007 (0.06) | 6.6e-11<br>1.2e-14<br>3.4e-14 | -0.77<br>-0.75<br>-0.79 | |