## [Peer Review File · Nature]

Manuscript Title: Regime shift in Arctic Ocean sea ice thickness

Reviewer Comments & Author Rebuttals

Reviewer Reports on the Initial Version:

Referees' comments:

Referee #1 (Remarks to the Author):

Summary

This manuscript presents an analysis of sea ice thickness and sea ice motion data that together suggest that a change in circulation patterns of Arctic sea ice has led to a shorter residence time, which in turn has led to a thinning of ice exported through Fram Strait. The authors first show evidence for a step-like change in the thickness distribution of sea ice that occurred in 2007 using data from upward looking sonars (ULS), which have been deployed nearly continuously in Fram Strait since 1990. Specifically, they draw attention to the change in position and shape of the primary mode of the thickness distribution that represents a thinning of level ice and a reduced influence of mechanical thickening. The authors then use daily gridded sea ice velocity data to compute back-trajectories of ice observed by the ULS and demonstrate a notable reduction in the time taken for ice to reach the Fram Strait after formation, with two step-like changes occurring in 2006 and 2007. Lastly, the authors develop a simple model of ice thickening due to thermodynamics growth and stochastic deformation events, which is able to reproduce the observed changes in the ice thickness distribution under the observed residence times. From this, the authors draw their primary conclusions that the abrupt changes in ice thickness in Fram Strait are a product of circulatory changes in the Arctic Ocean that result a reduced residence time for ice grown in the marginal seas.

Overall, the approach is novel and the manuscript is well written, with adequate description of the data and methods and good use of figures to show results. However, I have two major concerns regarding over simplification of the stochastic mechanical thickening model and a lack of any mention of uncertainty of measurements and derived results. The latter can be likely be addressed without significant change to the manuscripts findings, but the choice of assumptions in the stochastic model may overstate the importance of residence time as a cause of the abrupt change observed in the ice thickness distribution. I provide more detail on both these areas of concern below.

Major Comments

1. Stochastic model assumptions

I have two concerns with the setup of the stochastic ice thickening model related to assumptions about the uniformity with which deformation events are assumed to occur. First, I am concerned by the assumption that the frequency of deformation events remains constant despite changes in the thickness distribution and drift rate of the ice. For example, Zhang et al (2012) note that although ridging will increase as thick ice is replaced with thin ice, this is more than offset by increased

divergence and less ice overall in the marginal seas.

My second concern relates to the the “proportionate growth” assumption (line 285), which fails to consider the tendency for thin ice to deform before thick ice (e.g., Zhang et al, 2012). I agree that a deformation event in thick ice is likely to result in a greater thickness increase than one in thin ice, but with no weighting scheme to account for the reduced likelihood of thick ice being deformed, the model may overestimate the tendency for the variance of the thickness distribution and fraction of thick ice to increase with residence time. As a result of these two simplifying assumptions, I believe the stochastic model may be overlooking an alternative explanation for the changes in the thickness distribution, whereby the reduced variance of the distribution is caused by the reduce range of ice thickness taking part in ridging events. That is, if there is less thick ice available to become ridged, the ridges will be smaller and the thickness distribution will become compressed.

2. Lack of discussion of measurement uncertainty

I was a little surprised to find no mention of measurement uncertainties in either the text or figures. I appreciate that the uncertainty in ice draft measurements is considerably less than the standard deviation in monthly means, as illustrated in Figure 1, but I would have expected to see some estimation of the uncertainties in back trajectories and the residence times derived from them. For example, Mahoney et al (2019) use a similar particle tracking approach based on the same gridded ice velocity data and represent trajectories in the form of plumes accounting for the velocity uncertainty, which is provided for every datapoint. A similar approach could be taken here to estimate the resulting uncertainty in residence time.

The good news is that I believe the regime-shift analysis appears to be rather robust. I have to confess that I was initially a little skeptical of the regime shifts identified in Figures 3 and 4, which did not appear visually convincing partly because the variance does not appear to remain constant throughout the timeseries. Nonetheless, after digitizing the timeseries data from Figure 3a I was able to reproduce very nearly the same shifts using Rodionov’s sequential test. I was particularly impressed with how robust the timing of the regime changes was even when I added significant random noise to the data.

Minor comments

Line 19 (and elsewhere):

I recommend avoiding use the term “uniform” to describe the post-2007 sea ice regime. If the ice were uniform, the thickness distribution would resemble a single spike. I understand the authors are looking for a simple way to label the different regimes before and after 2007, but I feel the term “uniform” is inaccurate for this purpose.

Line 70 (and elsewhere):

I recommend replacing the phrase “dynamical growth” with “dynamical thickening” here and elsewhere in the manuscript. In the context of sea ice, the term “growth” implies the creation of new ice through freezing, whereas dynamical processes redistribute ice that has already formed.

Line 81: As per my comment above, I recommend replacing “grown” with “thickened”

<b?\>References cited in this review

Mahoney, A. R., J. K. Hutchings, H. Eicken, and C. Haas (2019), Changes in the Thickness and Circulation of Multiyear Ice in the Beaufort Gyre Determined From Pseudo-Lagrangian Methods from 2003–2015, *Journal of Geophysical Research: Oceans*, 124(8), 5618-5633 10.1029/2018jc014911.

Zhang, J., R. Lindsay, A. Schweiger, and I. Rigor (2012), Recent changes in the dynamic properties of declining Arctic sea ice: A model study, *Geophys Res Lett*, 39(20) <https://doi.org/10.1029/2012GL053545>.

Referee #2 (Remarks to the Author):

Evaluation: minor revision

According to the request, I reviewed the manuscript, titled "Regime shift in Arctic Ocean sea ice thickness" by Sumata et al. Basically, I enjoyed the reading throughout the manuscript since it is clearly written for its original points. I would like to highly evaluate their work and also recommend the manuscript to be published from Nature after some minor changes addressed below.

General comments:

The knowledge of ice thickness at the Fram Strait, originally collected by Norwegian Polar Institute, is comprehensively combined with the multifarious datasets that are publicly available like sea ice motion and sea ice concentration. I also find the analyses presented are really good, e.g., backward tracking of ice drift for its birth place and the detection of regime shifting. The authors newly found that the regime shift in thickness of ice floes from deformed to undeformed tendencies occurred around 2007. Then, they explain the occurrence by the shortened residence time of ice floes in the central Arctic and consequently reduced chance of deformation. With regard to the proportionate analytical model (Fig. 5), I have some questions about the choice of some constants like "b". The authors should more quantitatively evaluate how much the choices could affect the conclusions.

Specific and minor comments:

line # 175: "correlates" you may need to give some statistic evidence to claim this. Just give a correlation coefficient or something after interpolating them along the consistent time series and making a direct comparison between them.

185: "only ice floes ... " -> "only new ice floes .."

219: I'm fine with the point that the accelerated ice drift caused the shortened residence time.

However, the mean sea-ice drift is largely governed by the synoptic scale atmospheric circulation as presented in Kwok et al (Ref # 31). In my understanding, the cited papers analyzed the data only before 2009. You may need to show how the SLP pattern has changed after the time of regime shift in 2007. You might also show the AO index how it varies in 2010s as Kwok et al did. Another way of estimating ice drift magnitude is to take the horizontal gradient of sea level pressure across TDS, i.e., $U_{ice} = 13,000 \times \text{cosec}(\text{latitude}) \times d(\text{SLP})/dx$. You can refer to the classical theory by Zubov (1945) (see the text book by P. Wadhams, Ref. # 28).

267: "thermodynamical equilibrium thickness" regarding this, I think that you have to cite more relevant publications. At least, please have the title of the literature translated in English. In my suggestion, you could refer to some classical empirical models (e.g., Anderson, 1961) with the concept of "cumulative time of freezing degree". That is, an accumulation of daily mean of air temp minus freezing point of sea water follows an asymptotic curve. With the estimate, you can also verify your assumption of the undeformed ice forming the modal peak, around 1.5 m?, for the recent decade. You might say like "ice floes are subject to the freezing air for XX days to be YY m thick".

290: Spell out what PI means.

299: I suggest to move here the sentence in Methods, starting with "Since the number of events has not .. " at line # 548.

320: "Xm" should use a subscript m.

536 (Eqn. 8): how much could the choice of b change your results? See the related comments above.

538: I did not get the random variables r_i and r_{ii} stand for in practice. Please try to have clearer explanation for many potential readers, maybe by showing some examples.

541: "ridging and/or rafting" I think the ratio of these two deformation processes (in frequency) determines the value of b, and makes some difference in the final results. Again, you should some relevant references for the choice of $b = 0.4$ that accounts for the frequency of the two processes.

561: To save time of review by your reviewers, please add DOI numbers if they are recently published. e.g., Kwok et al. (2013) got the number: 10.1002/jgrc.20191.

Figure 1: Perhaps readers wants to quickly get where the location of the mooring from materials in the main text. Please consider to merge Fig. S1 into Fig. 1.

Figure 2: Please add units for variables in (b) and (c)

Figure 3: "thin grey lines .. " I cannot discern the lines. it looks just tiny dots; a typo in caption: "navy - while" -> "navy - white".

Figure S1: Again, please consider to merge this into Figure 1.

Reference:

Anderson, DL, 1961, Growth rate of sea ice , J. Glaciology, 3(30), 1170-1172.

Author Rebuttals to Initial Comments:

Response to the reviewers' comments.

We sincerely appreciate the anonymous reviewers for their thorough reviews, constructive comments, and suggestions on our manuscript. The reviews helped to improve robustness of the results and to polish the concept of the proportionate ice thickening model significantly. The suggestions and comments have been closely followed and revisions have been made accordingly.

In the following, the review comments are shown in *blue italic fonts* while our replies are in black roman font. The line number in our reply (shown in **red**) refers to those in the revised manuscript. References to the papers that appeared in the text are provided at the end of this response.

Referees' comments:

Referee #1 (Remarks to the Author):

Summary

This manuscript presents an analysis of sea ice thickness and sea ice motion data that together suggest that a change in circulation patterns of Arctic sea ice has led to a shorter residence time, which in turn has led to a thinning of ice exported through Fram Strait. The authors first show evidence for a step-like change in the thickness distribution of sea ice that occurred in 2007 using data from upward looking sonars (ULS), which have been deployed nearly continuously in Fram Strait since 1990. Specifically, they draw attention to the change in position and shape of the primary mode of the thickness distribution that represents a thinning of level ice and a reduced influence of mechanical thickening. The authors then use daily gridded sea ice velocity data to compute back-trajectories of ice observed by the ULS and demonstrate a notable reduction in the time taken for ice to reach the Fram Strait after formation, with two step-like changes occurring in 2006 and 2007. Lastly, the authors develop a simple model of ice thickening due to thermodynamics growth and stochastic deformation events, which is able to reproduce the observed changes in the ice thickness distribution under the observed residence times. From this, the authors draw their primary conclusions that the abrupt changes in ice thickness in Fram Strait are a product of circulatory changes in the Arctic Ocean that result a reduced residence time for ice grown in the marginal seas.

Overall, the approach is novel and the manuscript is well written, with adequate description of the data and methods and good use of figures to show results. However, I have two major concerns regarding over simplification of the stochastic mechanical thickening model and a lack of any mention of uncertainty of measurements and derived results. The latter can be likely be addressed without significant change to the manuscripts findings, but the choice of assumptions in the stochastic model may overstate the importance of residence time as a cause of the abrupt change observed in the ice thickness distribution. I provide more detail on both these areas of concern below.

First of all, we sincerely appreciate thorough assessments and thoughtful comments on our manuscript. To address these comments, we revised the stochastic model formulation and revised the manuscript accordingly.

Major Comments

1. Stochastic model assumptions

I have two concerns with the setup of the stochastic ice thickening model related to assumptions about the uniformity with which deformation events are assumed to occur.

We appreciate the thoughtful comments on the manuscript. Regarding the proportionate ice thickening model, we revised the model in order to take into account the two points that Reviewer #

1 raised below. In the revised model, 1) the frequency of deformation events depends on thickness distribution and 2) thinner ice has larger likelihood of deformation. These two characteristics are implemented in the areal probability of dynamical thickening. Please find our descriptions below.

First, I am concerned by the assumption that the frequency of deformation events remains constant despite changes in the thickness distribution and drift rate of the ice. For example, Zhang et al (2012) note that although ridging will increase as thick ice is replaced with thin ice, this is more than offset by increased divergence and less ice overall in the marginal seas.

We agree that this is an important point to be addressed when we consider the effect of ice thickness distribution on the deformation process. We revised our formulation in order to take this effect into account.

In the revised model, the areal probability of dynamical thickening, α , is given by a function of ice thickness, i.e., thinner ice is more likely to be deformed than thicker ice when a dynamical event occurs. This term also stands for the effect of long-term ice thickness changes and increase of mobility of ice. To implement this feature, we give the areal probability of dynamical thickening occurrence, α , as a function of ice thickness: the areal probability is inversely proportional to the stochastic ice thickness X_i ,

$$\alpha_i(X_i) = \frac{8}{X_i + 1} [\%]. \quad (\text{R1})$$

The formula indicates that ice of 1 m thickness experiences dynamical thickening at 4% areal probability, while 3 m thick ice at 2% probability. The new formulation means that when an ice pack experiences a convergence event (e.g., during a storm passage), 4% area of the ice is supposed to experience dynamical thickening if the ice is 1 m thick, while 2% area experiences thickening if the ice is 3 m thick. Once a thickening occurs, the thickness gain is governed by proportionate thickening process (i.e., thicker ice has larger likelihood to gain greater thickness) with a stochasticity. The latter is suggested by both modelling and observations (Melling&Riedel, 1996; Hopkins, 1994). This process includes both deformation of thicker ice types themselves and accumulation of deformed thinner ice forming ridges at the edges of thicker ice floes.

This formulation is based on an observational estimate of areal fraction of dynamical thickening during a deformation event (Itkin et al., 2018). According to the high-resolution survey of a single dynamical event, thickening occurred in 4% of the survey area, which was covered by ice with a mean thickness of 1.45 m. Referring to this number, we defined Eq. (R1) so that 1 m thick ice gives 4% probability. Implementation of this scheme is described in the revised manuscript (lines 651 - 661)

Since the functional form of this dependency is not known so far, we tested different formulations for $\alpha_i(X_i)$ in addition to the fixed case ($\alpha = 4\%$): a) $\alpha_i(X_i) = -0.25X_i + 3$ (weak linear dependency), b) $\alpha_i(X_i) = -0.5X_i + 4$ (strong linear dependency), and c) $\alpha_i(X_i) = 4 / (X_i + 1)$ (inversely proportional). The thickness distributions from the different probability formulations after 120 dynamical events are summarized in Figure R1. In general, introduction of the dependency slows down the formation of very thick ice compared to the fixed case, while the resultant thickness distributions have a similar lognormal shape when we apply 4 – 2% probability for 1 – 4 m thickness range. We also mention that this formula needs further evaluation by future observations that addresses the relation between areal probability of deformation and ice thickness (since we are not aware of direct observations of this beyond the Itkin et al. 2018 work).

Figure R1. Probability density functions of ice thickness distribution after 120 dynamical events for different areal probability formulations. The same thermodynamic growth term with (Eq. (9) in Methods) is applied.

In general, implementations of the thickness-dependent probability of thickening occurrence decelerate dynamical formation of thicker ice, and hence slows down the shift of the modal peak and increase of the variance (Figure R2). This process, however, does not affect the qualitative results and hence the conclusion of the manuscript, i.e., 1) the process forms a lognormal ice thickness distribution with large m , and 2) the longer residence time results in a smaller modal peak height and larger variance of the thickness distribution. In the revised manuscript, we estimated realistic ranges of each parameter and applied these in the model (lines 633 – 650, 657 – 671). The result shows that the model can demonstrate the changes of the ice thickness distribution from before to after the regime shift with parameters in a realistic range, by which we argue that the model captures the essence of the dynamical thickening process (lines 716 – 719).

We would like to emphasize that the entire nature of the model is conceptual, with its major purpose to demonstrate that the shape of the observed ice thickness distribution can be described by a lognormal distribution. Its functional form, in turn, can be derived from a relatively simple yet physically sensible combination of dynamical and thermodynamical terms with a few parameters based on realistic assumptions and observations. Moreover, we note that results of our study are not dependent on the actual values of the model parameters, but rather uses the major characteristics of the distribution obtained directly from the fit to the observational data.

Figure R2. Comparison of ice thickness distribution between fixed and thickness-dependent areal probability of thickening occurrence.

My second concern relates to the the “proportionate growth” assumption (line 285), which fails to consider the tendency for thin ice to deform before thick ice (e.g., Zhang et al, 2012). I agree that a deformation event in thick ice is likely to result in a greater thickness increase than one in thin ice, but with no weighting scheme to account for the reduced likelihood of thick ice being deformed, the model may overestimate the tendency for the variance of the thickness distribution and fraction of thick ice to increase with residence time.

This point is now taken into account by the implementation of the areal probability of the dynamical thickening. In the revised model, thicker fraction of ice has a smaller likelihood of dynamical thickening occurrence, while the thicker ice has larger likelihood to create larger ridges once a deformation occurs. The reduced likelihood of thick ice being deformed results in a reduction of the increase in variance of the thickness distributions (compared to the constant likelihood, Fig. R2). The application of parameters with realistic ranges, however, suggest that the proportionate thickening process describes the essence of the dynamical thickening process, together with the precedence of thinner ice deformation. The implementation of the scheme strengthened robustness of the model, for which we appreciate the thoughtful comments from Reviewer #1.

*As a result of these two simplifying assumptions, I believe the stochastic model may be overlooking an alternative explanation for the changes in the thickness distribution, whereby **the reduced variance of the distribution is caused by the reduce range of ice thickness taking part in ridging events. That is, if there is less thick ice available to become ridged, the ridges will be smaller and the thickness distribution will become compressed.***

The model applied now in the revision of our manuscript has successfully demonstrated the process that Reviewer #1 mentioned (part of the bold-faced sentences in the above paragraph). The model takes into account “**the reduced range of ice thickness taking part in ridging events**” after the regime shift, i.e., thinner ice has smaller potential to produce very thick ridged ice and the larger likelihood of thin ice deformation could not compensate this. As a consequence, “**there is less thick ice available to become ridged**”, and then it resulted in the situation after 2007, i.e., “**the ridges will be smaller and the thickness distribution will become compressed.**”. Please note that the model demonstrated that this process is explained by the shorter residence time, if we assume a proportionate dynamical thickening together with the precedence of thinner ice deformation.

2. Lack of discussion of measurement uncertainty

I was a little surprised to find no mention of measurement uncertainties in either the text or figures. I appreciate that the uncertainty in ice draft measurements is considerably less than the standard deviation in monthly means, as illustrated in Figure 1, but I would have expected to see some estimation of the uncertainties in back trajectories and the residence times derived from them. For example, Mahoney et al (2019) use a similar particle tracking approach based on the same gridded ice velocity data and represent trajectories in the form of plumes accounting for the velocity uncertainty, which is provided for every datapoint. A similar approach could be taken here to estimate the resulting uncertainty in residence time.

Thank you for pointing out an important issue that should have been addressed in the manuscript. Regarding the ice thickness distributions, we provided the description of the error assessment in Methods as follows (lines 487 – 511),

“The uncertainties on the estimates of monthly fractions of ice thicker than a threshold and position of the modal ice peak were assessed numerically using a moving block bootstrap approach (Künsch, 1989). Bootstrapping is a family of resampling techniques used for deriving uncertainties on various complex estimators for large data sets and employs random sampling with replacement (Efron and

Tibshirani, 1993) The presence of autocorrelation in ice draft series from IPS (ULSs deployed after 2006) suggests the use of the moving block bootstrap approach (Künsch, 1989). The method splits the original monthly series of ice drafts $O(10^6)$ samples of length N into $N-K+1$ overlapping blocks of length K each. The block length was set to 30 samples that approximately corresponds to a distance of 10 m covered by ice traveling at 0.3 knots. It roughly corresponds to a lower limit of a horizontal spatial scale of ice ridges. For ES300 instruments (ULSs deployed until 2005) with a lower sampling rate of 240 seconds the ordinary bootstrapping was used.

At each of M steps of bootstrap sampling, N/K blocks are drawn at random, with replacement, from the constructed set of $N-K+1$ blocks making a new bootstrap data sample for the month. A Gaussian noise of $N(0,1)$ is further added to the data to account for measurement uncertainty. The mean and standard deviation of the fractions of thicker ice and modal ice peak position are then calculated directly from the M estimates derived on each step of the procedure. Results suggest that the monthly coefficient of variation or a ratio of the standard deviation of the estimate to its mean, for IPS data varies within 1-3% for both fractions, being on average lower (1-2 %) for the fraction of ice > 4 m thick. For ES300 the coefficient of variation is slightly higher of about 4(6)% for the fractions of ice thicker than 4(5) m. The same applies to position of the modal peak that shows coefficient of variation of 0-3% being typically closer to 0 for IPS data. It suggests that the selected bin width is large enough to accommodate uncertainties related with the approach and the data. For the ES300 data the monthly STD of the modal peak position is higher up to 30 cm and the coefficient of variation to lie within 9%. We therefore postulate that the inferred uncertainties are far too low to have any significant influence on the results of the shift detection analysis. “

Uncertainties of daily positions of the backward trajectories are assessed by comparisons with buoy tracks obtained from the International Arctic Buoy Program (IAPB) (Rigor et al., 2002). We used 83 buoy tracks that arrived in Fram Strait from 2000 to 2018, and calculated corresponding pseudo buoy tracks backward in time (Figure R3). Figure R4 shows a summary of the comparisons.

Figure R3. Comparisons between IAPB buoy tracks (left) and the backward trajectory calculation (right). Python Matplotlib basemap toolkit is used to plot the map.

The comparisons show that the mean error of the daily pseudo buoy position (red line in Figure R4) can be reasonably approximated by a linear function of backtracking days (dashed black line in Fig. R4), $\text{Error} = 50 + (\text{backtracking days}) / 2$ [km]. We apply this empirical formula as an error of the daily position of the backward trajectories from 0 to 300 backtracking days, while apply a constant error of 200 km after 300 days. The latter is because the number of validation data significantly

reduces after 300 days and the error estimate is strongly influenced by few exceptional cases. We suggest this is a reasonable approximation since the large fraction of the error samples (blue dots in Fig. R4) still resides within 200 km distance after 300 days.

Figure R4. Error of the daily positions of backward trajectories as a function of backtracking days. The blue dots indicate distance between daily IABP buoy positions and corresponding pseudo buoy positions, which are used to quantify the error of the backward trajectory calculation (taken from from suppl. material of Sumata et al. 2022).

Figure R5 shows examples of backward trajectories of pseudo ice floes starting in the Fram Strait together with their error estimates. The spatial extent of the uncertainty area (gray shade) surrounding the ice formation locations (orange dots) indicates that the possible error of the trajectory calculation does not affect the estimated area of ice formation significantly (i.e., the error shade around the orange dots are contained in the polygon which we defined in Fig. 3b and c).

Origins and pathways of sea ice arrived in Fram Strait in November 2006

Figure R5. Trajectories of pseudo ice floes that arrived at the Fram Strait section in November 2006. The left panel shows ice floes formed in 2002, while the right panel shows those formed in 2003. The orange dots show locations of sea ice formation, the thin black lines show trajectories, and gray shade shows error of the trajectories.

The background white-blue graduation shows September sea ice concentration in 2002 (left) and 2003 (right). Python Matplotlib basemap toolkit is used to plot the map.

We agree that it could be informative to display trajectories together with their uncertainty as Mahoney et al. (2019) demonstrated. In Figure 3b and c in our manuscript, however, it is visually not possible to show the uncertainty shade since the number of the trajectories in Fig. 3 is $O(10^3)$ (Fig 3b: 1237 trajectories, Fig. 3c: 1006 trajectories). Although each trajectory is not distinguishable in Figs 3b and c, our purpose is to show that the mean and typical sea ice pathways (demonstrated as the cloud of the trajectories, i.e., shape of thereof) have not significantly changed before and after 2007 (lines 123 – 124). For this reason we find that visualizing the uncertainty of every trajectory is visually not informative for this purpose, but we have now included descriptions of the uncertainty estimates of the trajectories in the revised manuscript (lines 577 – 587).

We estimated the uncertainty of the residence time of ice floes that appeared in Fram Strait by the standard deviation of residence time of 8 pseudo ice floes (Figure R6). For the backward trajectory calculation, we started with these 8 pseudo ice floes across the western part of the Fram Strait section (78.8°N , from 0° to 10°W) at the same time. The backward calculations start on the 15th of each month from 1990 to 2019. The initial distance between neighboring pseudo floes is 21 – 42 km, and the largest distance between the floes is 210 km. After a couple to several years of backward advection, the floes are tracked back to the sea ice formation areas. The resultant residence time, however, significantly differs between the floes regardless of their initial proximity, as a consequence of convergent ice motion in the central Arctic and the large interannual variation in the spatial pattern of the sea ice edge in the Alaskan and Siberian sectors (see e.g., ice concentration in Figure R5). For example, Figure R5 shows trajectories of ice floes that arrived in Fram Strait on 15th November, 2006. Four trajectories end up in 2002 (panel a), while other four trajectories end up in 2003 (panel b). Such a nature of the residence time variation is consistent with our findings from in situ observations in Fram Strait. A variety of sea ice with seemingly different ages observed within a small distance ($O(10 - 100)$ km). For these reasons we assume the standard deviation of the residence time of the floes give a good measure of the uncertainty of the mean residence time of ice floes that appeared in Fram Strait at a certain point in time. We revised Figure 3a and describe the uncertainty estimates in Methods (lines 588 – 592). The uncertainty (std. dev.) is now also shown in Extended Data Fig. 2.

Figure R6. Mean residence time of sea ice that appeared in Fram Strait. The standard deviation is calculated from the residence time of 8 pseudo ice floes that arrived in Fram Strait at the same time (revised Figure 3a).

The good news is that I believe the regime-shift analysis appears to be rather robust. I have to confess that I was initially a little skeptical of the regime shifts identified in Figures 3 and 4, which did not appear visually convincing partly because the variance does not appear to remain constant throughout the timeseries. Nonetheless, after digitizing the timeseries data from Figure 3a I was

able to reproduce very nearly the same shifts using Rodionov's sequential test. I was particularly impressed with how robust the timing of the regime changes was even when I added significant random noise to the data.

We appreciate Reviewer #1 for the independent test on the detected shift with random noise, which strengthened the robustness of the result.

Minor comments

Line 19 (and elsewhere):

I recommend avoiding use the term "uniform" to describe the post-2007 sea ice regime. If the ice were uniform, the thickness distribution would resemble a single spike. I understand the authors are looking for a simple way to label the different regimes before and after 2007, but I feel the term "uniform" is inaccurate for this purpose.

Thank you for the suggestion. We discussed among the authors and concluded to use 'more uniform' to describe the post-2007 sea ice regime, since 'less deformed' is also misleading. The manuscript is now revised accordingly.

Line 70 (and elsewhere):

I recommend replacing the phrase "dynamical growth" with "dynamical thickening" here and elsewhere in the manuscript. In the context of sea ice, the term "growth" implies the creation of new ice through freezing, whereas dynamical processes redistribute ice that has already formed.

Thank you for the suggestion. We agree to the point brought up here.

We revised the manuscript accordingly and use 'proportionate thickening' instead of 'proportionate growth', to explain our formulation.

Line 81: As per my comment above, I recommend replacing "grown" with "thickened"

We revised the texts accordingly.

References cited in this review

*Mahoney, A. R., J. K. Hutchings, H. Eicken, and C. Haas (2019), Changes in the Thickness and Circulation of Multiyear Ice in the Beaufort Gyre Determined From Pseudo-Lagrangian Methods from 2003–2015, Journal of Geophysical Research:Oceans, 124(8), 5618-5633
10.1029/2018jc014911.*

*Zhang, J., R. Lindsay, A. Schweiger, and I. Rigor (2012), Recent changes in the dynamic properties of declining Arctic sea ice: A model study, Geophys Res Lett, 39(20)
<https://doi.org/10.1029/2012GL053545>.*

Referee #2 (Remarks to the Author):

Evaluation: minor revision

According to the request, I reviewed the manuscript, titled "Regime shift in Arctic Ocean sea ice thickness" by Sumata et al. Basically, I enjoyed the reading throughout the manuscript since it is clearly written for its original points. I would like to highly evaluate their work and also recommend the manuscript to be published from Nature after some minor changes addressed below.

General comments:

The knowledge of ice thickness at the Fram Strait, originally collected by Norwegian Polar Institute, is comprehensively combined with the multifarious datasets that are publicly available like sea ice motion and sea ice concentration. I also find the analyses presented are really good, e.g., backward tracking of ice drift for its birth place and the detection of regime shifting. The authors newly found that the regime shift in thickness of ice floes from deformed to undeformed tendencies occurred around 2007. Then, they explain the occurrence by the shortened residence time of ice floes in the central Arctic and consequently reduced chance of deformation. With regard to the proportionate analytical model (Fig. 5), I have some questions about the choice of some constants like "b". The authors should more quantitatively evaluate how much the choices could affect the conclusions.

We sincerely appreciate the thoughtful comments and suggestions by Reviewer #2. Regarding the last point in the above paragraph, we provided explanations how the parameters are chosen and described the model's sensitivity to the choices (see also our responses to Reviewer #1). Please find our detailed responses below.

Specific and minor comments:

line # 175: "correlates" you may need to give some statistic evidence to claim this. Just give a correlation coefficient or something after interpolating them along the consistent time series and making a direct comparison between them.

Thank you for pointing this out. We added correlation coefficient at the end of this sentence, "($r = 0.65$ in Alaskan sector, 0.73 in Siberian sector, ice concentration leads 1 year)". (lines 128 – 129).

185: "only ice floes ... " -> "only new ice floes .."

We revised the text accordingly.

219: I'm fine with the point that the accelerated ice drift caused the shortened residence time. However, the mean sea-ice drift is largely governed by the synoptic scale atmospheric circulation as presented in Kwok et al (Ref # 31). In my understanding, the cited papers analyzed the data only before 2009. You may need to show how the SLP pattern has changed after the time of regime shift in 2007. You might also show the AO index how it varies in 2010s as Kwok et al did. Another way of estimating ice drift magnitude is to take the horizontal gradient of sea level pressure across TDS, i.e., $U_{ice} = 13,000 \times \text{cosec}(\text{latitude}) \times d(\text{SLP})/dx$. You can refer to the classical theory by Zubov (1945) (see the text book by P. Wadhams, Ref. # 28).

Thank you for the comments and suggestions on the relation between sea ice drift and atmospheric wind forcing. We have not addressed this point in the manuscript because we did not find clear relations between the timing of the regime shift and change of the atmospheric wind forcing in our analyses. We have analyzed sea level pressure (SLP) pattern and 10 m wind over the Arctic Ocean from ERA5. Although the interannual variation of TPD speed has been strongly governed by the wind forcing (Figure R7 and Figure 4d), we didn't find any clear change of wind forcing at the time of the regime shift. The sequential t-test also did not detect any regime shift in the wind time series for the entire analyzed period (1990 – 2019) shown in Fig. R7. In addition, increase of the wind speed (26% in TPD area after 2007) could not fully explain the acceleration of the TPD (37%), even if we ignore the timing of the shift. These result suggests that the sudden acceleration of the TPD at the time of the regime shift (Fig. 4d) was not directly caused by the change of wind forcing, but rather associated with the change of sea ice conditions affecting sea ice mobility, as pointed by Olason and Notz (2014). For these reasons, we speculate that the TPD acceleration at the time of the regime shift is driven by change of the ice conditions triggered by the summer ice reduction in the ice formation areas, while the enhanced wind forcing has contributed to the TPD acceleration. To

address the latter point, we now included Fig.R7 and Fig. R8 together in Extended Data Fig. 7 in the manuscript. Our discussion about these points appears in the manuscript as follows (lines 226 – 229),

“The younger ice is thin, weakly linked, and features ridges with more shallow keels, and hence more prone to wind forcing pushing the ice toward the Atlantic sector of the Arctic. This process has accelerated the TPD from 2007 onwards (Fig. 4d), while an enhanced wind forcing after 2007 may also have contributed to the acceleration of the TPD (Extended Data Fig. 7).”.

Figure R7. Time series of annual mean 10 m wind averaged in the three polygons shown in Fig. R8. The mean wind speed in the rectangular box C (representing Transpolar Drift area) is a component of wind vector parallel to the major axis of the rectangular box C (positive wind speed orients to Fram Strait). (included in Extended Data Fig. 7)

Figure R8. Difference of sea level pressure (SLP) and wind field between two periods, 1990 – 2006 and 2007 – 2019, in (a) summer (from June to November) and (b) winter (from December to May). Polygons shown in the panels indicate sea ice formation areas, A: Alaskan sector and B: Siberian sector, and C: the Transpolar Drift stream. Python Basemap toolkit is used to plot the map. (included in Extended data Fig. 7)

267: "thermodynamical equilibrium thickness" regarding this, I think that you have to cite more relevant publications. At least, please have the title of the literature translated in English. In my suggestion, you could refer to some classical empirical models (e.g., Anderson, 1961) with the

concept of "cumulative time of freezing degree". That is, an accumulation of daily mean of air temp minus freezing point of sea water follows an asymptotic curve. With the estimate, you can also verify your assumption of the undeformed ice forming the modal peak, around 1.5 m?, for the recent decade. You might say like "ice floes are subject to the freezing air for XX days to be YY m thick".

Thank you for the suggestion. We revised the manuscript to address the points brought up here. Regarding the thermodynamic process of sea ice, we now cite Leppäranta (1993) in the revised manuscript. We also added an application for Anderson's law to describe the initial condition of the stochastic ice thickening model. The initial condition is set to 1 m in the revised manuscript, which roughly corresponds to the thickness of new ice, after three month of its initial formation (based on Anderson's freezing-degree days law, with an assumption of ice surface temperature of 253K) (lines 695 – 697). This is obviously not the thermodynamical equilibrium thickness, while the model includes thermodynamic ice growth together with the dynamical thickening starting from the initial thickness. We describe this process now in the revised manuscript (lines 672 – 697). The title of the cited paper is now translated in English.

290: Spell out what PI means.

Thank you for pointing this out. Π is a product operator. We revised the manuscript accordingly.

299: I suggest to move here the sentence in Methods, starting with "Since the number of events has not .." at line # 548.

Thank you for the suggestion. We revised the manuscript accordingly.

320: "Xm" should use a subscript m.

Thank you for pointing out the typo. We revised the text.

536 (Eqn. 8): how much could the choice of b change your results? See the related comments above.

For the first attempt to develop the proportionate (dynamical) ice thickening model, we referred to detailed field observations by Itkin et al. (2018). Sensitivity of the model results on the choice of the proportionate thickening constant b is now described in Methods (lines 701 – 709). Please also find our response below.

538: I did not get the random variables r_i and r_{ii} stand for in practice. Please try to have clearer explanation for many potential readers, maybe by showing some examples.

Thank you for the suggestion.

In the revised manuscript, we now provide an explanation of r_i , the stochastic thickening increment and introduced the areal probability of dynamical thickening, α , instead of r_{ii} .

We also explain what the formulation means as follows,

“This formula indicates that when a dynamical event occurs, α % area of pack ice experiences dynamical thickening (ridging/rafting), while the rest $(1 - \alpha)$ % remains unchanged (areal stochasticity). Thickness gain, $b \cdot r_i$, in the dynamical thickening area, is also a stochastic variable: the possible maximum gain is b while the minimum is 0 (r_i is random, $0 \leq r_i < 1$) (thickening stochasticity).” (lines 628 – 632).

Also, please find relevant explanation in the following responses.

541: "ridging and/or rafting" I think the ratio of these two deformation processes (in frequency) determines the value of b , and makes some difference in the final results. Again, you should some relevant references for the choice of $b = 0.4$ that accounts for the frequency of the two processes.

Thank you for the comments. In the model formulation, the proportionate thickening constant b gives the upper bound of ice thickness that can be gained by one dynamical event in ridging/rafting area, while the frequency (probability) of occurrence of dynamical thickening is governed by the areal probability of dynamical thickening given by α_i . The introduction of α implicitly takes into account the different probability of ridging and rafting, i.e., probability of thickening occurrence depends on the baseline thickness. As described below, however, the relation between the ridging/rafting ratio and the baseline thickness is not known so far, we employed a simple formulation that represents both process together.

The thickness gain is a stochastic variable, given by $b \cdot r$, where r ranges $0 \leq r < 1$ with constant probability, i.e., the minimum gain is zero while the maximum possible gain is b .

The choice of $b = 0.4$ indicates that, for example, sea ice in the ridging/rafting area gains 0.4 m thickness at maximum (0.2 m on average) when the ice is initially 1 m thick, while it gains 1.2 m thickness at maximum (0.6 m on average) when initially ice is 3 m thick. Though it is difficult to estimate value of b from in-situ observations so far, Itkin et al. (2018) provided an indication of a rough estimate of this value from a description of a single dynamical thickening event. They analyzed spatial details of sea ice deformation and concurrent thickening (5 m \times 5 m resolution, covering a 9 km² area) obtained from airborne laser scanner surveys just before and right after a storm event over Arctic sea ice. According to their analysis, the change of sea ice freeboard in a converging area is 0.07 m (0.36 m \rightarrow 0.43 m) on average, corresponding to 0.58 m thickening of ice by ridging/rafting (assuming freeboard to thickness ratio = 8.35, Vinje and Finnekåsa (1986)). Since the mean ice thickness in their survey area was 1.45 m, the gain is $b = 0.58 / 1.45 = 0.4$, if we assume a proportionate thickening process. Though this number comes from one dynamical event, we applied the value as a first attempt to develop the proportionate ice thickening model (in absence of other such data to develop the model further).

It should be mentioned that $b = 0.4$ is an areal average obtained from Itkin et al. (2018), while we applied this as the upper bound of b . This is because the probability function of thickening increment, $b \cdot r$, is not known so far, and we assumed constant probability between 0 and b . A larger value of b , e.g., $b = 0.8$, deforms resultant ice thickness distribution to a bi-modal form, since the thickness gained by the constant probability causes too much thickening near the upper bound and hence produce another peak apart from the modal thickness. If we apply a smaller value, e.g., $b = 0.2$, we generally obtain a similar ice thickness distribution with a lognormal shape, but it requires a larger number of dynamical events, m , to reach the same modal thickness and variance. We described these points (how to derive the first estimate of b and the sensitivity of the result on the choices of b) now in the revised manuscript (lines 633 – 650 and 699 - 707).

We also mentioned that the value of the proportionate thickening constant, b , together with its probability function should be further examined with future observations (lines 707 – 709).

The frequency (probability) of dynamical thickening in the formulation is governed by the areal probability of the dynamical thickening. In our initial formulation (in the first version of our manuscript), the areal probability was fixed to 2%, i.e., dynamical thickening occurs at 2% probability (interpreted as 2% area of ice-covered area experiences dynamical thickening at a dynamical event such as a storm passage). This occurs regardless of the ice thickness before the event. In the revised manuscript, however, we modified the formulation in order to take into account dependence of ice strength on thickness (following the suggestion from Reviewer #1, this also implicitly takes into account different probability of ridging and rafting). In the new formulation, the areal probability of dynamical thickening, α_i , is given by a function that inversely proportional to the thickness (α_i given by a linear decrease with thickness gives the similar result, see our replies

to Reviewer #1 above). This represents the fact that thinner ice is more prone to the compressive force and is easily ridged or rafted while thicker ice can resist the force to a certain extent. In the new formation, for example, 1 m thickness ice experiences dynamical thickening at 4% areal probability, while 3 m ice at 2% probability. To estimate realistic range of α_i , we referred to an observational estimate of areal fraction of dynamical thickening (Itkin et al., 2018): According to their estimate, dynamical thickening occurred in 4% of the survey area, where the mean thickness of the area before the thickening event was 1.45 m. We selected these values for the model with simplification, i.e., thickening occurs at 4% areal fraction when ice is 1 m, while the probability of the areal fraction is reduced with increased ice thickness.

To provide the more detailed explanations of the model and to fulfill the word limit requested by the editor, we moved a large part of the proportionate ice thickening model description to Methods (lines 601 - 719).

561: To save time of review by your reviewers, please add DOI numbers if they are recently published. e.g., Kwok et al. (2013) got the number: 10.1002/jgrc.20191.

Sorry for the inconvenience. Now we added doi numbers to all references, when they are available.

Figure 1: Perhaps readers wants to quickly get where the location of the mooring from materials in the main text. Please consider to merge Fig. S1 into Fig. 1.

Thank you for the suggestion. We merged Fig. S1 into Fig. 1.

Figure 2: Please add units for variables in (b) and (c)

The selected properties of the ice thickness distributions are unitless, since the distributions are derived from the number of samples in each bin [counts/month] divided by the total number of the samples [counts/month]. (described in lines 471 – 472).

Figure 3: "thin grey lines .. " I cannot discern the lines. it looks just tiny dots; a typo in caption: "navy - while" -> "navy – white".

Thank you for pointing out this issue. Since we plot $O(10^3)$ trajectories (thin grey lines) in Fig. 3b and c, it is actually not possible to discern each lines. Since our purpose here is to show that the mean and typical pathways of sea ice that arrived in Fram Strait have not significantly changed before and after the regime shift, it is not necessary to discern each trajectory. To address this point, we revised the figure legend and now describe the trajectories as ‘the gray cloud’.

Figure S1: Again, please consider to merge this into Figure 1.

We have merged Fig. S1 into Fig. 1.

Reference:

Anderson, DL, 1961, Growth rate of sea ice , J. Glaciology, 3(30), 1170-1172.

Finally, we again appreciate the review comments and suggestions on our manuscript, which significantly helped to polish the idea of the stochastic model and to improve its presentation.

Best regards,
Hiroshi Sumata, on behalf of all authors

References cited in this response

- Efron, B. and R. J. Tibshirani, An Introduction to the Bootstrap, *New York: Chapman & Hall, software*, 456 pp., (1993).
- Hopkins M.A. On the ridging of intact lead ice. *J. Geophys. Res.*, 99, 16351-16360, <https://doi.org/10.1029/94jc00996> (1994).
- Itkin, P., G. Spreen, S. M. Hvidegaard, H. Skourup, J. Wilkinson, S. Gerland, and M. A. Granskog, Contribution of deformation to sea ice mass balance: A case study from an N-ICE2015 storm. *Geophys. Res. Lett.*, **45**, 789-796. <https://doi.org/10.1002/2017GL076056>. (2018)
- Künsch, H. R., The jackknife and the bootstrap for general stationary observations, *The annals of Statistics*, **17** (3), 1217 – 1241, <https://doi.org/10.1214/aos/1176347265> (1989).
- Leppäranta, M., A review of analytical models of sea-ice growth, *Atmosphere-Ocean*, **31**:1, 123 – 138, <https://doi.org/10.1080/07055900.1993.9649465> (1993)
- Mahoney, A. R., J. K. Hutchings, H. Eicken, and C. Haas, Changes in the Thickness and Circulation of Multiyear Ice in the Beaufort Gyre Determined From Pseudo-Lagrangian Methods from 2003–2015, *Journal of Geophysical Research:Oceans*, **124** (8), 5618-5633, <https://doi.org/10.1029/2018JC014911> (2019).
- Melling H, Riedel D.A. Development of seasonal pack ice in the Beaufort Sea during the winter of 1991-1992: A view from below, *J. Geophys. Res.*, 101, N C5, 11975-11991. <https://agupubs.onlinelibrary.wiley.com/doi/epdf/10.1029/96JC00284> (1996).
- Olason, E. and D. Notz, Drivers of variability in Arctic sea-ice drift speed, *J. Geophys. Res. Oceans*, **119**, 5755 – 5755, <https://doi.org/10.1002/2014JC009897> (2014).
- Rigor, I. G., Wallace, J. M. and Colony, R. L. Response of Sea Ice to the Arctic Oscillation. *J. Clim.* **15**, 2648 – 2663, [https://doi.org/10.1175/1520-0442\(2002\)015%3C2648:ROSITT%3E2.0.CO;2](https://doi.org/10.1175/1520-0442(2002)015%3C2648:ROSITT%3E2.0.CO;2) (2002).
- Sumata, H., L. de-Steur, S. Gerland, D. V. Divine, and O. Pavlova, Unprecedented decline of Arctic sea ice outflow in 2018, *Nature communications*, **13**:1747, <https://doi.org/10.1038/41467-022-29470-7> (2022).
- Vinje, T., and Finnekåsa, Ø. The ice transport through Fram Strait, *Norsk Polarinstitutt Skr.*, **186**, 1 – 41, <http://hdl.handle.net/11250/173520> (1986).

Reviewer Reports on the First Revision:

Referees' comments:

Referee #1 (Remarks to the Author):

Regime shift in Arctic Ocean sea ice thickness
by Hiroshi Sumata and others

Submitted to Nature

2nd Review: November 10, 2022

Summary

The authors have done an outstanding job of thoughtfully responding to all my comments and those of the other reviewer. I am particularly impressed with the introduction of a thickness dependency in their stochastic ridging model, which completely satisfies any residual doubts I had about the role of shortened residence time in the observed changes in the ice thickness distribution at Fram Strait. I also appreciate the additional details regarding uncertainties in the ice thickness measurements and pseudo floe trajectories, but I have some concerns that the authors may be underestimating the uncertainty in the latter. I describe this in more detail below with a suggestion for how to estimate the errors in the absence of independent buoys data.

However, I have some remaining comments related to the use of IABP data for assessing the accuracy of computed back trajectories of ice arriving at Fram Strait. I list these as “major” comments below, but I believe they can both be addressed with the inclusion of some additional caveats in the text.

Major comment:

Use of IABP data may over-estimate accuracy of computed back trajectories

The comparison between drift tracks of IABP buoys and pseudo floes looks extremely favorable, but unfortunately this does represent an independent validation since IABP data are already incorporated into the NSIDC ice velocity dataset. Using this comparison as the basis for estimating positional error in the computed back trajectories is therefore likely to overestimate the accuracy of all such back trajectories. It would be much better to compare the computed trajectories with the drift tracks of buoys not included in the gridded ice velocities (for example, I again refer the authors to the work of Mahoney et al, 2019, as cited in my last review).

Additionally, none of the IABP buoy tracks shown in Figure R3 of the rebuttal document originate in the source regions shown in Figure 3 of main text. And few of buoy trajectories come close in duration to the residence times calculated for the pseudo floes. As a result, the authors are not able to use their comparison to accurately determine the positional errors much beyond 300 days before arrival at Fram Strait. They therefore truncate the time-dependent growth in uncertainty at 200 km after 300 days on the basis that the uncertainty for most individual trajectories is below this level.

However, for the reasons described above, the positional uncertainties shown in Figure R4 should not be taken as representative for all computed back trajectories and I therefore this decision may further under-estimate the uncertainty in the computed back trajectories.

The ideal solution to this problem would be to find an alternative dataset of GPS-tracked buoys that are not already included in the IABP archive and which were deployed near the source regions.

Unfortunately, I expect there are few, if any, such buoys. I therefore suggest two alternative approaches to addressing my concern:

Suggestion 1:

My first suggestion involves the greatest amount of additional work, but parallels the boot-strap approach already applied to estimating uncertainty in the ice draft data and I believe would result in a more representative and complete assessment of positional uncertainty over time than a comparison with IABP drift tracks. By repeatedly calculating each back trajectory a large enough number of times, and randomly sampling each time the specified error estimates at each grid cell, the authors could generate a range of possible trajectories that would define the effective uncertainty in position over time. This is in fact the same approach used by Mahoney et al (2019), as referenced in my earlier review. The authors are correct that visualizing the results of this approach for thousands of trajectories would be challenging, but this does not mean the approach cannot be implemented on this many trajectories. Specifically, Mahoney et al used this approach to create over 10,000 “pseudoplumes” (computed every day for 12 years at 4 mooring locations), with each plume representing an ensemble of 10,000 individual trajectories. In any case, it may be possible to derive a representative assessment of the time-dependent uncertainty using a small number of trajectories. More detailed information on the per-grid cell error estimates is described in the documentation that accompanies the ice velocity dataset at NSIDC.

Suggestion 2:

My alternative suggestion is a lot simpler and would be to linearly extrapolate the mean positional error, rather than truncating it at 200 km after 300 days. In Figure R4, the mean error appears to continue linearly out to at least 500 days so I find difficult to justify the truncation at 300 days. In addition, I would recommend that the authors acknowledge that the comparison with IABP buoy tracks does not represent an independent validation and therefore likely underestimates the positional uncertainty.

Referee #2 (Remarks to the Author):

After my review of the revised manuscript, by the authors (Sumata et al.), I find they did very good job for every concerns and suggestions that I previously made.

According to my previous suggestions they explored the possibility of changes in surface pressure pattern that could affect the regime shift in ice drift. Also, they sufficiently improved the discussions for the dependency on the choice of stochastic parameters, e.g. b. Furthermore, they have adopted the concept of accumulated days of freezing temperature as per my past suggestion.

I am satisfied with their responses and cannot find any more complaint for them, with this revised version. I would like to say ok for the publication.

Author Rebuttals to First Revision:

Response to the reviewers' comments.

We sincerely appreciate the reviewers for their reviews and suggestions on our revised manuscript. We further revised the manuscript based on the suggestions by reviewer #1. In the following, the reviewer comments are shown in *blue italic font* while our replies are in black roman font. The line number in our reply (shown in **red**) refers to those in the revised manuscript (attached to this current revision). References to the papers that appear in the text are provided at the end of this response letter.

Referees' comments:

Referee #1 (Remarks to the Author):

*Regime shift in Arctic Ocean sea ice thickness
by Hiroshi Sumata and others*

Submitted to Nature

2nd Review: November 10, 2022

Summary

The authors have done an outstanding job of thoughtfully responding to all my comments and those of the other reviewer. I am particularly impressed with the introduction of a thickness dependency in their stochastic ridging model, which completely satisfies any residual doubts I had about the role of shortened residence time in the observed changes in the ice thickness distribution at Fram Strait. I also appreciate the additional details regarding uncertainties in the ice thickness measurements and pseudo floe trajectories, but I have some concerns that the authors may be underestimating the uncertainty in the latter. I describe this in more detail below with a suggestion for how to estimate the errors in the absence of independent buoys data.

However, I have some remaining comments related to the use of IABP data for assessing the accuracy of computed back trajectories of ice arriving at Fram Strait. I list these as "major" comments below, but I believe they can both be addressed with the inclusion of some additional caveats in the text.

Thank you for the further assessments and suggestions on our manuscript. Please find our replies and corresponding revision below.

Major comment:

Use of IABP data may over-estimate accuracy of computed back trajectories

The comparison between drift tracks of IABP buoys and pseudo floes looks extremely favorable, but unfortunately this does represent an independent validation since IABP data are already incorporated into the NSIDC ice velocity dataset. Using this comparison as the basis for estimating positional error in the computed back trajectories is therefore likely to overestimate the accuracy of all such back trajectories. It would be much better to compare the computed trajectories with the drift tracks of buoys not included in the gridded ice velocities (for example, I again refer the authors to the work of Mahoney et al, 2019, as cited in my last review).

Additionally, none of the IABP buoy tracks shown in Figure R3 of the rebuttal document originate in the source regions shown in Figure 3 of main text. And few of buoy trajectories come close in

duration to the residence times calculated for the pseudo floes. As a result, the authors are not able to use their comparison to accurately determine the positional errors much beyond 300 days before arrival at Fram Strait. They therefore truncate the time-dependent growth in uncertainty at 200 km after 300 days on the basis that the uncertainty for most individual trajectories is below this level. However, for the reasons described above, the positional uncertainties shown in Figure R4 should not be taken as representative for all computed back trajectories and I therefore this decision may further under-estimate the uncertainty in the computed back trajectories.

The ideal solution to this problem would be to find an alternative dataset of GPS-tracked buoys that are not already included in the IABP archive and which were deployed near the source regions. Unfortunately, I expect there are few, if any, such buoys. I therefore suggest two alternative approaches to addressing my concern:

Thank you for pointing out this issue.

We agree the point that IABP buoy tracks cannot be regarded as completely independent validation data for the trajectory calculation, since they were already taken into account NSIDCv4 ice drift vectors.

And as reviewer #1 pointed out, however, it is difficult to find sufficient number of GPS-tracked buoys for validation that deployed in the ice formation areas in the Siberian sector, since they belong to the Russian exclusive economic zone and in recent times also lack of ice in the area. These situation makes it difficult to deploy drifters and hence it is known that this side of the Arctic has less coverage of buoys. For this reason we revised the manuscript based on the second suggestion by reviewer #1 below.

Suggestion 1:

My first suggestion involves the greatest amount of additional work, but parallels the boot-strap approach already applied to estimating uncertainty in the ice draft data and I believe would result in a more representative and complete assessment of positional uncertainty over time than a comparison with IABP drift tracks. By repeatedly calculating each back trajectory a large enough number of times, and randomly sampling each time the specified error estimates at each grid cell, the authors could generate a range of possible trajectories that would define the effective uncertainty in position over time. This is in fact the same approach used by Mahoney et al (2019), as referenced in my earlier review. The authors are correct that visualizing the results of this approach for thousands of trajectories would be challenging, but this does not mean the approach cannot be implemented on this many trajectories. Specifically, Mahoney et al used this approach to create over 10,000 “pseudoplumes” (computed every day for 12 years at 4 mooring locations), with each plume representing an ensemble of 10,000 individual trajectories. In any case, it may be possible to derive a representative assessment of the time-dependent uncertainty using a small number of trajectories. More detailed information on the per-grid cell error estimates is described in the documentation that accompanies the ice velocity dataset at NSIDC.

Suggestion 2:

My alternative suggestion is a lot simpler and would be to linearly extrapolate the mean positional error, rather than truncating it at 200 km after 300 days. In Figure R4, the mean error appears to continue linearly out to at least 500 days so I find difficult to justify the truncation at 300 days. In addition, I would recommend that the authors acknowledge that the comparison with IABP buoy tracks does not represent an independent validation and therefore likely underestimates the positional uncertainty.

We appreciate these suggestions and took the second option described above. Thus, in the revised manuscript we describe the concern raised by Reviewer #1 of the use of IABP buoy tracks for validation and have now extended the application of the empirical formula of the error estimate until 500 days (lines 580 - 584). We also describe the caveat that our error estimate could underestimate the uncertainty of the trajectories due to the lack of independent validation data. At

the same time, we refer to Mahoney et al. (2019) who showed that comparisons between non-IABP buoy tracks and pseudo buoy tracks derived from NSIDCv4 with error estimates by a bootstrap method demonstrated that pseudo tracks remain largely parallel to the corresponding buoys and that the error does not monotonically increase over time (lines 584 - 588). Since we applied the same ice drift product to derive the trajectories as Mahoney et al. (2019) used, their thorough error assessments supports the use of the trajectory method. Given that the error analysis now indicates an error of order of 300 km, compared to the region of origin (order of 1500 km), this has no influence on the conclusions that have been drawn in the paper.

Referee #2 (Remarks to the Author):

After my review of the revised manuscript, by the authors (Sumata et al.), I find they did very good job for every concerns and suggestions that I previously made.

According to my previous suggestions they explored the possibility of changes in surface pressure pattern that could affect the regime shift in ice drift. Also, they sufficiently improved the discussions for the dependency on the choice of stochastic parameters, e.g. b. Furthermore, they have adopted the concept of accumulated days of freezing temperature as per my past suggestion.

I am satisfied with their responses and cannot find any more complaint for them, with this revised version. I would like to say ok for the publication.

We sincerely appreciate the positive assessment of our revised manuscript, and still like to thank for the suggestions by Reviewer #2 that helped to improve the paper.

Best regards,
Hiroshi Sumata, on behalf of all authors

Reference

Mahoney, A. R. et al., Changes in the Thickness and Circulation of Multiyear Ice in the Beaufort Gyre determined from Pseudo-Lagrangian Methods from 2003-2015. *J. Geophys. Res. Oceans*, 124, 5618-5633. <https://doi.org/10.1029/2018JC014911> (2019).